# Role of Secreted Frizzled-Related Protein 1 in Early Breast Carcinogenesis and Breast Cancer Aggressiveness

**DOI:** 10.3390/cancers15082251

**Published:** 2023-04-12

**Authors:** Alisson Clemenceau, Aurélie Lacouture, Juliette Bherer, Geneviève Ouellette, Annick Michaud, Étienne Audet-Walsh, Caroline Diorio, Francine Durocher

**Affiliations:** 1Department of Molecular Medicine, Faculty of Medicine, Laval University, Quebec, QC G1V 0A6, Canada; alisson.clemenceau.1@ulaval.ca (A.C.);; 2Cancer Research Centre, CHU de Quebec Research Centre, Quebec, QC G1V 4G2, Canada; 3Department of Social and Preventive Medicine, Faculty of Medicine, Laval University, Quebec, QC G1V 0A6, Canada

**Keywords:** breast cancer, Secreted Frizzled-Related Protein 1, *SFRP1*, lobular involution, organoids, triple-negative breast cancer, atypical ductal hyperplasia, estrogen receptor, *ESR1*

## Abstract

**Simple Summary:**

*Secreted Frizzled-Related Protein 1* (*SFRP1*) expression is decreased during breast cancer progression and inversely associated with breast tissue age-related lobular involution in women. In this study, we have investigated the potential causal role of the decrease in *SFRP1* expression in early breast carcinogenesis. In order to do so, we characterized mammary epithelial cells from both nulliparous and multiparous mice in organoid culture ex vivo, and modulated *SFRP1* expression in breast non-tumoral and tumoral cell lines from multiple molecular subtypes. Our results support the hypothesis that a lack of *SFRP1* could have a causal role in early breast carcinogenesis.

**Abstract:**

A human transcriptome array on ERα-positive breast cancer continuum of risk identified *Secreted Frizzled-Related Protein 1* (*SFRP1*) as decreased during breast cancer progression. In addition, SFRP1 was inversely associated with breast tissue age-related lobular involution, and differentially regulated in women with regard to their parity status and the presence of microcalcifications. The causal role of *SFRP1* in breast carcinogenesis remains, nevertheless, not well understood. In this study, we characterized mammary epithelial cells from both nulliparous and multiparous mice in organoid culture ex vivo, in the presence of estradiol (E2) and/or hydroxyapatite microcalcifications (HA). Furthermore, we have modulated *SFRP1* expression in breast cancer cell lines, including the MCF10A series, and investigated their tumoral properties. We observed that organoids obtained from multiparous mice were resistant to E2 treatment, while organoids obtained from nulliparous mice developed the luminal phenotype associated with a lower ratio between *Sfrp1* and *Esr1* expression. The decrease in *SFRP1* expression in MCF10A and MCF10AT1 cell lines increased their tumorigenic properties in vitro. On the other hand, the overexpression of *SFRP1* in MCF10DCIS, MCF10CA1a, and MCF7 reduced their aggressiveness. Our results support the hypothesis that a lack of *SFRP1* could have a causal role in early breast carcinogenesis.

## 1. Introduction

Recent decades have seen the advent of systematic mammographic screening in women older than 50 years old in developed countries, as well as the development of targeted therapies such as tamoxifen, a selective estrogen receptor modulator antagonizing the estrogen receptor-alpha (ERα) in the breast. To personalize breast cancer care, tumors can nowadays be classified by clinical subtypes, assessed by immunohistochemistry and described by the St. Gallen expert consensus, i.e., luminal breast cancer expressing ERα and/or the progesterone receptor (PR), overexpressing (luminal B)—or not (luminal A/B)—the human epidermal growth factor receptor 2 (HER2) and/or having (luminal B)—or not (luminal A)—a high proliferative index (Ki-67) [1,2,3]. In addition to these two molecular subtypes, the St. Gallen expert consensus also classifies HER2-positive breast tumors, negative for both ERα and PR but overexpressing HER2, as well as triple-negative breast cancers (TNBC), which express neither ERα nor PR and do not overexpress HER2 [2,3]. On the other hand, tumors can also be classified by transcriptome analyses, defining the tumor intrinsic molecular subtype. Multiple classifications were developed since the first one was proposed by Perou et al. in 2000, among which there is a diagnostic method termed, Prediction Analysis of Microarray 50 (PAM50), which differentiates five molecular subtypes based on 50 gene expression profiles, i.e., luminal A (high expression of *ESR1*, *GATA3,* and *XBP1*), luminal B (high expression of *ESR1* and *HER2*), HER2-enriched (high expression of *HER2* and *EGFR*), basal-like (high expression of *CDK5*, *CDK17*, *laminin,* and *FABP7*) and normal-like (high expression of *CD36* and *GPX3*) [4,5].

If ERα is a crucial marker of breast cancer diagnosis and a well-known therapeutic target against the disease, it is important to keep in mind that, during a woman’s lifetime, ERα expression in breast tissue does not remain constant, even in a physiological context. Indeed, ERα is more expressed during breast morphogenesis periods, including embryogenesis and puberty [6,7,8]. It is also more expressed during the follicular phase of the menstrual cycle, while it is not expressed during the luteal phase in adult women, and it could also be increased after age-related lobular involution completion [9,10,11]. In addition, the expression of ERα is decreased in parous women compared to nulliparous women [12,13]. A better comprehension of such physiological mechanisms of regulation could help not only to prevent early ERα-positive and -negative carcinogenesis, but also to identify new therapeutic targets in order to fight against this societal burden. This is even more crucial given that multiparity remains a well-known yet not well understood protective factor against breast cancer development in women [14,15,16]. Indeed, some studies reported that parity was associated with a decrease in luminal breast cancer risk specifically, corroborating the importance of studying the physiological regulation of ERα in the breast [14,17,18,19].

In order to better understand the early ERα-positive carcinogenesis, our group previously performed a human transcriptome array on normal tissue, atypical ductal hyperplasia (ADH), ERα-positive ductal carcinoma in situ (DCIS) and ERα-positive invasive ductal carcinoma (IDC) [20]. Among the genes differentially expressed between non-tumoral and invasive lesions, *Secreted Frizzled-Related Protein 1* (*SFRP1*) was identified [20]. At the molecular level, SFRP1 is a Wnt signaling pathway antagonist largely associated with tissue homeostasis regulation and, consequently, with carcinogenesis [20,21]. In addition, a few studies in mice depleted of *Sfrp1* demonstrated that it negatively regulates mammary branching and the expression of ERα, raising the question of the causal role the lack of SFRP1 plays in luminal breast cancer development in women [22].

In another study, our group demonstrated that SFRP1 was largely expressed in non-involuted breast epithelial tissue, while it dramatically decreased after perimenopausal lobular involution completion in nulliparous women. Furthermore, we observed that SFRP1 expression was also associated with the presence of microcalcifications in breast tissue, once more in nulliparous women specifically [23]. However, in the absence of well-characterized models of lobular involution and microcalcification development in both nulliparous and multiparous mammalians, the knowledge regarding SFRP1 regulation through pregnancy and hydroxyapatite (HA) microcalcifications remains unsubstantial. It is even more important to decipher these potential cross-regulations, as both nulliparity and clusters of microcalcifications are associated with an increase in breast cancer risk in women [14,15,16,24].

Hence, we hypothesize that the lack of *SFRP1* in non-tumoral breast tissue results in mammary gland branching in nulliparous mammalians, and in the acquisition of a malignant phenotype, increasing *ESR1* expression and, consequently, sensitizing breast cells to estradiol (E2) proliferative and migratory effects. We also anticipate that an increase in *SFRP1* expression in breast cancer cell lines should decrease cell viability and reduce *ESR1* expression, at least in luminal breast cancer cell lines.

In the present study, we first analyzed public datasets to compare *SFRP1* expression and promoter methylation among breast non-tumoral and tumoral tissues. Thereafter, we compared the correlation between *SFRP1* and biomarkers of breast cancer molecular subtyping, such as *ESR1*, *PGR,* and *HER2* in both breast non-tumoral and tumoral tissues. We then compared the expression of *Sfrp1* in nulliparous and multiparous mice mammary glands cultured ex vivo in the presence, or not, of E2 and/or HA crystals. We also reported on the phenotypic characteristics of the organoids depending on parity, the presence of E2 and/or HA, and *SFRP1* expression. In addition, we analyzed the causal role of *SFRP1* modulations of expression by lentiviral transduction on early breast carcinogenesis and breast cancer aggressiveness in vitro. We also investigated the existence of an E2-dependent modulation of *SFRP1* expression, as well as a potential *SFRP1*-dependent response to E2 in breast cell lines. Finally, we examined the association between *SFRP1* expression in tumoral tissue and patients’ outcomes after stratification for ERα status or breast cancer molecular subtypes.

## 2. Materials and Methods

### 2.1. Analyses of Public Datasets

*SFRP1* expression across breast non-tumoral, tumoral, and metastatic tissues was obtained by using the TNMplot public dataset (https://tnmplot.com/, accessed on 15 April 2021) [25]. *SFRP1* expression analyses regarding breast cancer degree of differentiation, molecular subtypes (assessed by immunohistochemistry), and grade were obtained from the GENT2 public dataset (http://gent2.appex.kr/gent2/, accessed on 15 April 2021) [26]. Co-expression analyses were obtained from the UCSC Xena visualization tool (https://xenabrowser.net/, accessed on 18 September 2021) on the TCGA Breast Cancer (BRCA) tumoral dataset (n = 1247) and on normal breast tissue (Benz 2020, n = 151). Breast cancer molecular subtypes in the TCGA BRCA tumoral dataset were assessed by the PAM50 diagnostic method. Survival curves were drawn using the KM-plot online tool (accessed on 15 April 2021) [27]. Overall survival (OS) and recurrence-free survival (RFS) were used as outcomes on ERα-positive (OS n = 754; RFS n = 2633), ERα-negative (OS n = 520; RFS n = 1190), luminal A (OS n = 794; RFS n = 2277), luminal B (OS n = 515; RFS n = 1491), basal (OS n = 404; RFS n = 846), and HER2 (OS n = 166; RFS n = 315) specific subgroups. ERα-positive and -negative subgroups have been defined following the immunohistochemistry status; the other 4 subgroups have been defined following the St. Gallen consensus [28]. Follow-up was censored at 180 months and the median of *SFRP1* expression was used to segregate high *SFRP1* expression from low *SFRP1* expression. Each molecular subtype pathological response to chemotherapy in association with *SFRP1* expression was obtained from the ROCplot.org database (ROCplot; http://www.rocplot.org/, accessed on 2 June 2021). The same probe set (202037_s_at), targeting *SFRP1,* was used for all previously cited analyses. PREdiction of Clinical Outcomes from Genomic Profiles (PRECOG; https://precog.stanford.edu/, accessed on 16 April 2021) [29] was used to determine the meta-Z score of the association between *SFRP1* expression and breast cancer overall survival (https://kmplot.com/analysis/, accessed on 15 April 2021) [27].

### 2.2. Mice Organoid Culture and Data Collection

Primary culture of mouse organoids was performed as previously detailed by Lacouture et al. [30]. Protocols were performed according to the Université Laval Research and Animal Ethics Committee’s guidelines and regulations. Briefly, C57BL/6 nulliparous and multiparous mice were sacrificed, and inguinal mammary glands were collected. Both mice and sampled mammary glands were weighed. One of the two inguinal glands has been fixed in formalin and embedded in paraffin for further analyses. The other gland was conserved on ice in complete Hank’s balanced salt solution (HBSS) before being cut, pooled with additional glands according to the subject’s parity status, and digested in 1X solution of Gentle Collagenase/Hyaluronidase (StemCell, Vancouver, BC, Canada) with 1X complete EpiCult-B mouse medium + 5% FBS (EpiCult basal medium (StemCell) + 10 ng/mL recombinant human EGF (StemCell) + 4 μg/mL heparin (Sigma-Aldrich Canada Co., Oakville, ON, Canada) + 100 U/mL penicillin and 100 μg/mL streptomycin) overnight at 37 °C. The mixture was then submitted to centrifugation, and the pellet resuspended in 1 mL of warm 0.25% trypsin-EDTA (Wisent) for 3 min. Thereafter, 4 mL of warm 0.25% trypsin-EDTA (Wisent, Québec, QC, Canada) was added, and the mixture was kept on ice for 1 h before adding 10 mL of 1X complete HBSS solution, followed by centrifugation at 350× *g* for 5 min. Supernatant was removed and 2 mL of warm dispase (5 U/mL; StemCell) and 0.1–1 mg/mL of DNAse I (Roche, Basel, Switzerland) were added. After homogenization by pipetting, 10 mL of cold 1X complete HBSS solution was added, and the suspension was filtered through a 40 μm cell strainer (Falcon, London, UK) before centrifugation at 350× *g* for 5 min. Cells were then counted, and a purification process was performed using the EasySep Mouse Epithelial Cell Enrichment kit II (StemCell) according to the manufacturer’s protocol. Thereafter, cells were plated for 1 h for differential plating (to avoid overgrowth of fibroblasts). Purified mammary epithelial cells were centrifuged at 350× *g* for 5 min and resuspended in complete EpiCult-B mouse medium (StemCell) + 5% FBS and 75% growth-factor-reduced Matrigel (Corning, Somerville, MA, USA) + 100 ng hydroxyapatite crystals (Sigma), or not, for a total of 30,000 cells by 40 μL droplet. Droplets were plated in 24-well plates with cold tips and then incubated upside down for 15 min at 37 °C. Subsequently, 3 mL of warm complete EpiCult-B mouse medium (StemCell) + 5% FBS at 37 °C with 5% CO_2_ was added. Medium was replaced 24 h later by serum-free complete EpiCult-B mouse medium with 10 nM EtOH or E2, and changed every 3 days. Pictures were taken every 3 days with the EVOS M5000 Imaging System (ThermoFisher Scientific, Waltham, MA, USA) and analyzed using ImageJ software.

### 2.3. Cell Lines

The embryonic kidney HEK293T cell line was used for transfection and lentivirus production, and was kindly provided by Dr. Stéphane Gobeil (CHU de Québec Research Centre). The HEK293T cell line was grown in DMEM medium supplemented with 10% fetal bovine serum on previously coated (5 mg/mL poly-L-lysine) petri dishes. All breast cell line characteristics and culture media used in the present study are summarized in Table 1. The MCF10A (ATCC^®^ CRL-10317™) breast cancer risk continuum obtained from a Caucasian 36-year-old woman was used to investigate the role of SFRP1 on early breast carcinogenesis, and was obtained from the Cell Resources Facility of the Barbara Ann Karmanos Cancer Institute, Wayne State University. The MCF10AT1 cell line, representing ADH, was derived from a stable transfection of MCF10A with mutant Ha-ras while MCF10DCIS.com, representing DCIS, was obtained from a xenograft of MCF10AT1 on an immunodeficient mouse. The MCF10CA1a and MCF7 (ATCC^®^ HTB-22™) cell lines were used to explore the impact of SFRP1 modulation of expression on IDC aggressiveness in vitro. For all cell lines, culture media were changed every 48 to 72 h after washing with 1% PBS. Cells were split by adding 1 mL trypsin once they reached 60 to 70% confluency. Cells were maintained at 37 °C and 5% CO2. The hydrocortisone, insulin, and EGF were purchased from Sigma-Aldrich (Sigma-Aldrich Canada Co., Oakville, ON, Canada). E2 was obtained from Steraloids (Newport, RI, USA). All other cell culture reagents were purchased from Wisent Inc. (Québec, QC, Canada). All cell lines were shown to be mycoplasma free.

### 2.4. Lentiviral Production and Cell Infection

Three lentiviruses, each containing a different shRNA targeting *SFRP1,* i.e., TRCN0000062169, TRCN0000062170, and negative control shScramble in pLKO.1-puro vector were used to knock down *SFRP1*. These shRNAs were graciously provided by Dr. Stéphane Gobeil from Laval University, Quebec City, Canada and are respectively referred to as sh62169, sh62170, and shScramble throughout the manuscript. One lentivirus containing an open reading frame (ORF) plasmid construct for the *SFRP1* gene (BRDN0000460838) in pLX_TRC304 vector was used to overexpress *SFRP1* in breast cell lines. This ORF, graciously provided by Dr. Mathieu Laplante from Laval University, Quebec City, Canada, is referred to as OxSFRP1 throughout the manuscript. Empty pLX_TRC304 vector was used as negative control. The transfection was done in HEK293T cells plated in 60 to 70% confluency in 10 cm petri dishes coated with 5 mg/mL poly-L-lysine for 30 min at 37 °C. Lentiviruses were assembled by co-transfecting HEK293T with 12 μg shRNAs or ORF, 10.8 μg pMD2.G (VSVG coding plasmid), and 1.2 μg psPAX2 (lentivirus packaging plasmid) in 350 μL LBS buffer (pH = 4). After 48 h incubation at 37 °C and 5% CO2, the media containing lentiviruses were collected and filtered using 0.45 μm pore size filters (MilliporeSigma, Darmstadt, Germany). The lentiviruses were aliquoted in 1.5 mL tubes and stored at −80 °C until further use. Subsequently, 1.5 × 10^6^ cells were plated in 10 cm petri dishes, and each infected with 2 mL of either shRNAs, shScramble, or OxSFRP1 viruses. One mL of medium and 3 µL polybrene (10 µg/mL) were also added to each petri dish, and cells were incubated overnight at 37 °C and 5% CO2. After 12 to 16 h of incubation, the media were changed after washing 3 times with 1X PBS. After 24 h, cells were split and fresh selective media, containing 2.5 to 22 µg/mL of blasticidin (Wisent Inc., Québec, QC, Canada) for OxSFRP1, and 1 µg/mL of puromycin (InvivoGen, San Diego, CA, USA) for shRNAs and shScramble, were added. Three independent infections were done on 3 different cryovials for each cell line and each condition (OxSFRP1, sh62169, sh62170, and shScramble) in technical triplicate.

### 2.5. 2D Proliferation Assays

Cell lines were automatically counted with the BIORAD TC20 automated cell counter (Bio-Rad, Hercules, CA, USA) and 10,000 cells were plated in 200 µL of fresh media in 96-well plates. After 24 h incubation, media were replaced by fresh media containing 10 nM E2 or ethanol (EtOH). After 72 h incubation at 37 °C and 5% CO2, the media were changed with fresh medium containing 10% alamarBlue^®^ (Thermo Fisher Scientific, Waltham, MA, USA). The plates were then incubated at 37 °C with 5% CO2 until the first appearance of pink coloration. The absorbance was measured at 570 nm and 585 nm wavelengths with the TECAN Infinite M1000 (TECAN, Männedorf, Switzerland). The experiments were performed in triplicate and repeated three times independently.

### 2.6. Spheroids Assays

Cell lines were automatically counted with the BIORAD TC20 automated cell counter (Bio-Rad, Hercules, CA, USA) and 10,000 cells were plated in 200 µL of fresh media containing 10 nM E2 or EtOH in Ultra-Low Attachment (ULA) 96-well plates (Corning, Somerville, MA, USA). Pictures were taken with the EVOS™ M5000 imaging system (Invitrogen™, Thermo Fisher Scientific, Waltham, MA, USA) every 24 h for 5 days. The spheroid size was measured with ImageJ, and the average size relative to the respective negative control at day 1 was calculated. The experiments were performed in triplicate and repeated three times independently.

### 2.7. Migration Assays

Cell lines were automatically counted with the BIORAD TC20 automated cell counter (Bio-Rad, Hercules, CA, USA) and 25,000 cells were plated in 100 µL of fresh media in 2 chamber wells (Ibidi GMBH, Martinsried, Planegg, Germany). Cells were then incubated at 37 °C with 5% CO2. After 24 h, the chambers were removed, and the culture media were replaced with fresh media containing 10 µM mitomycin C to stop cell proliferation and 10 nM E2 or EtOH. Pictures were taken with the EVOS™ M5000 imaging system (Invitrogen™, Thermo Fisher Scientific, Waltham, MA, USA) immediately (0 h) and every 6 to 12 h until the gap between the cells was filled. The wound area was measured with ImageJ and the percentage of wound area was calculated relative to the wound area at 0 h of migration. The experiments were performed in triplicate and repeated three times independently.

### 2.8. RNA Extraction, Purification, and Reverse Transcription

Total RNA from organoids and cell lines was isolated with the Qiagen RNeasy mini kit (Qiagen, Hilden, Germany) following the manufacturer’s instructions, using 30 µL of warmed (37 °C) RNAse-free water for elution. Thereafter, 2 to 3 µg of total RNA were diluted in RNAse-free water for a final volume of 10 µL. Thereafter, 1 µL of dNTP (10 mM), 1 µL oligo dT (50 µM), and 1 µL random hexamers (50 µM) were added to the mixture before incubation at 65 °C in a dry bath for 5 min, followed by 1 min on ice. Subsequently, 1 µL of SuperScript IV Reverse Transcriptase, 4 µL of 5X Buffer, 1 µL RNAse out and 1 µL DTT (100 mM) were added to the mixtures and incubated for 10 min at room temperature followed by 15 min at 50 °C and 10 min at 80 °C in a dry bath. All reagents were purchased from Thermo Fisher Scientific (Waltham, MA, USA). Purification was performed with the QIAquick PCR Purification Kit (Qiagen, Hilden, Germany) following the manufacturer’s instructions, using 20 to 30 µL of warmed (37 °C) RNAse-free water for elution.

### 2.9. Quantitative Real Time Polymerase Chain Reaction (qPCR)

Oligo primers (Appendix A) were designed by using the primer-BLAST tool from the National Center for Biotechnology Information with the following conditions: the expected amplicon size was between 80 and 300 base pairs, the percentages of GC for both forward and reverse primers were ~50%, and their melting temperatures ~60 °C. The designed primers spanned an exon–exon junction and were specific to each of the targeted gene isoforms when possible. The reactions were plated in triplicate in 96-well plates for a final volume of 20 µL, composed of 50% (10 µL) PowerSYBR^®^ Green PCR Master Mix (Applied Biosystems, Waltham, MA, USA), 20 ng of cDNA, 2 µL of oligo mix (5 µM each primer), and DNAse-free water. Thereafter, the qPCRs were performed in the StepOne Plus Real-Time PCR System (Applied Biosystems, Waltham, MA, USA). The runs were started by 10 min at 95 °C followed by 40 cycles of 15 s at 95 °C followed by 1 min at 60 °C. Melt curves were performed in order to confirm the melting temperatures of the amplicons. Data calculation and normalization were performed using the second-derivative and double-correction methods. Three housekeeping genes, i.e., *Glyceraldehyde-3-phosphate dehydrogenase* (*GAPDH*), *Hypoxantine-guanine phosphoribosyltransferase* (*HPRT1*), and *Mitochondrial membrane ATP synthase* (*ATP5O*) for human cell lines, and 2 housekeeping genes, i.e., *Pumilio homolog 1* (*Pum1*) and *TATA binding protein 1* (*Tbp1*) for mouse epithelial cells, were quantified. DNAse-free water was used for no-template control for each oligo primer pair in order to detect potential contamination.

### 2.10. Statistical Analyses

All analyses described were performed with RStudio v1.2.5033 (RStudio Team (2019), RStudio: Integrated Development for R, RStudio, Inc., Boston, MA, USA, URL http://www.rstudio.com/ (accessed on 1 November 2021)). Comparisons between the average mouse weight and average weight of the mammary glands (n = 57) were performed using the unpaired two-sided Student’s *t*-test and using the *t* test R function from the stats package, version 4.1.2. For ex vivo and in vitro experiments (n = 3), comparisons of means between the two groups were performed by one-sided Mann–Whitney testing with the R function wilcox.test from the stats package, version 4.1.2. In both figures and the main text, ex vivo and in vitro results are presented as mean ± standard deviation. All experiments were performed in technical triplicate and performed three times independently. *p*-value lower than 0.05 was considered significant.

## 3. Results

### 3.1. SFRP1 Expression and Co-Expression Pattern across Breast Cancer Tissues

As already demonstrated in previous work, *SFRP1* expression decreases with the progression of breast lesions [20]. This was corroborated by the TNMplot.com public dataset [25] analyses, which demonstrated that *SFRP1* expression was lower in tumoral breast tissue (fold change (FC) = 0.20) compared to normal breast tissue, and lower in metastatic (FC = 0.77) compared to breast tumors (Kruskal-Wallis *p*-value < 0.0001; Figure 1A). Interestingly, in the GENT2 platform [26], *SFRP1* expression was higher in poorly differentiated breast tumors compared to moderately (log2 FC = 0.90; *p*-value < 0.05) and well-differentiated breast tumors (log2 FC = 2.49; *p*-value = 0.12; Appendix A). In addition, *SFRP1* expression was higher in grade 3 tumors compared to grade 2 (log2 FC = 0.86; *p*-value < 0.001) and grade 1 (log2 FC = 0.57; *p*-value < 0.01) tumors (Appendix A). *SFRP1* expression was also higher in triple-negative compared to luminal (log2 FC = 3.14; *p*-value < 0.001) and HER2 (log2 FC = 2.99; *p*-value < 0.001) breast tumors and in basal compared to luminal (log2 FC = 2.96; *p*-value < 0.001) and HER2 (log2 FC = 2.82; *p*-value < 0.001) breast tumors (Figure 1B).

From a mechanistic point of view, *SFRP1* decreased expression in tumor tissue compared to normal breast tissue seemed to be mediated by its promoter, methylation (*p*-value < 0.0001; Appendix A). By stratifying the TCGA cohort by the breast cancer molecular subtypes, we noticed that the *SFRP1* promoter was more methylated in luminal lesions compared to normal tissue (*p*-value < 0.0001; Appendix A), which was coherent with *SFRP1* decreased expression in luminal lesions compared to normal samples. However, it remained surprising that the *SFRP1* promoter was also more methylated in TNBC compared to in normal breast tissue (*p*-value < 0.0001; Appendix A), while the *SFRP1* level of expression was not different in TNBC compared to non-tumoral breast tissue. Co-expression analyses with the Xena online tool on the Benz 2020 non-tumoral breast tissue dataset showed that *SFRP1* expression was positively correlated with *ESR1* (rho = 0.16; *p*-value = 0.05), *PGR* (rho = 0.35; *p*-value = 1.7 × 10^−5^), and *HER2* (rho = 0.64; *p*-value = 4.0 × 10^−18^) expression, while it was negatively correlated with *ESR2* expression (rho = −0.28; *p*-value = 6.9 × 10^−4^). Corroborating our previous study, which demonstrated that *SFRP1* expression in non-tumoral breast tissue was inversely associated with age-lobular involution, we saw that *SFRP1* expression in non-tumoral tissue from the Benz 2020 dataset was positively correlated with the number of terminal ductal lobular units (TDLUs; rho = 0.53; *p*-value = 5.2 × 10^−12^; Figure 1C). On the other hand, in the BRCA invasive ductal carcinoma dataset, *SFRP1* expression was negatively correlated with *ESR1* (rho = −0.45; *p*-value = 2.6 × 10^−82^), *PGR* (rho = −0.13; *p*-value = 4.1 × 10^−6^), and *HER2* (rho = −0.29; *p*-value = 4.0 × 10^−25^) expression, while it was positively correlated with *ESR2* (rho = 0.47; *p*-value = 5.7 × 10^−67^; Figure 1D). 

To summarize, *SFRP1*, both at the mRNA and protein levels, was negatively correlated with the number of TDLUs, while positively correlated with *ESR1*, *PGR,* and *HER2* expression in non-tumoral breast tissue. On the other hand, *SFRP1* expression becomes negatively correlated with *ESR1*, *PGR,* and *HER2*, in tumoral breast tissue. In addition, the decrease in *SFRP1* expression in luminal breast cancer tissue seems induced by an increase in *SFRP1* promoter methylation. These interesting results prompted us to explore the reasons and consequences of such a “switch” between the positive correlations observed in non-tumoral tissue, and the negative correlations observed in breast tumoral tissue.

### 3.2. Sfrp1 Expression in Mice Organoids and Its Association with Organoid Phenotype

As we previously observed a difference in SFRP1 expression between nulliparous and parous women after perimenopausal lobular involution [23], and in the absence of well-characterized in vivo and in vitro models of lobular involution in both nulliparous and multiparous groups, we decided to develop ex vivo models obtained from nulliparous and multiparous mice. A total of 57 inguinal mice mammary glands were sampled, 28 from nulliparous mice and 29 from multiparous mice. The average weight in the nulliparous group was 25.6 ± 3.5 g, while the average weight in multiparous group was 34.3 ± 6.2 g (*p*-value = 2.6 × 10^−7^). The average age was also significantly different between both groups, i.e., 235.1 ± 56.9 days in the nulliparous group compared to 319.4 ± 48.8 days in the multiparous group (*p*-value = 1.4 × 10^−6^). The average mammary gland weight was higher in multiparous (0.4 ± 0.2 g) than in nulliparous (0.2 ± 0.07 g; *p*-value = 3.8 × 10^−6^; Appendix A) mice. Both weight (r = 0.9; *p*-value < 2.2 × 10^−16^; Appendix A) and age (r = 0.6; *p*-value = 5.3 × 10^−6^; Appendix A) were strongly correlated with mice mammary gland weight. We then compared mice mammary gland weight according to parity status after adjustment for age. We observed that mice mammary gland weight was still significantly higher in multiparous (adjusted mean = 0.4 ± 0.03 g) than in nulliparous (adjusted mean = 0.2 ± 0.03 g) mice (*p*-value = 0.004) meaning that mice mammary gland weight is associated with the parity status, independently from age.

To attest that mice ex vivo models could be a good model for a lobular involution study, we first performed hematoxylin-eosin (H&E) staining on formalin-fixed paraffin-embedded mice mammary gland tissues (Appendix A–F). A mammary gland obtained from a multiparous mouse 48 h post weaning showed many glandular structures, dilated ducts and acini-containing secretions (Appendix A). On the other hand, 25 days post weaning (Appendix A), we can only observe some residual flat acini without secretory activity. Finally, mammary glands from nulliparous mice are devoid of lobulo-alveolar structure in the mammary fat pad (Appendix A).

We previously observed that SFRP1 expression was different among nulliparous and parous women, in premenopausal compared to postmenopausal women, as well as in the presence compared to in the absence of breast microcalcifications. Along the same lines, we decided to mimic these conditions ex vivo by cultivating organoids obtained from nulliparous and multiparous mice, in the presence or in the absence of E2 and/or HA crystals [23].

*Sfrp1* expression was similar between organoids obtained from nulliparous and multiparous mice and grown in EtOH condition (Figure 2A). Treatment with E2 induced a 1.9 ± 0.6-fold decrease in nulliparous, and a 2.1 ± 0.1-fold decrease in multiparous mice, in *Sfrp1* expression compared to their respective controls grown in EtOH condition (all *p*-values < 0.05; Figure 2A). In addition, treatment with EtOH + HA induced a 2.0 ± 1.1-fold increase in *Sfrp1* expression in organoids obtained from multiparous mice, compared to those obtained from nulliparous mice (*p*-value = 0.06; Figure 2A). These results suggest that the regulation of *Sfrp1* expression in mammary epithelial cells is dependent on the parity status in mice. Moreover, *Esr1* expression was 3.1 ± 0.2-fold higher in organoids obtained from nulliparous mice than in organoids obtained from multiparous mice in EtOH condition (*p*-value < 0.05; Figure 2B), thus suggesting a higher sensitivity to estrogens in nulliparous mice compared to multiparous mice. This was corroborated by the measurement of *Pgr* expression in the organoids following E2 treatment. Indeed, while a 25.1 ± 13.1-fold increase in *Pgr* expression was observed in organoids obtained from nulliparous mice following treatment with E2, only a 5.6 ± 1.3-fold increase was observed in organoids obtained from multiparous mice compared to their respective controls grown in EtOH condition (all *p*-values < 0.05; Figure 2C). Furthermore, in organoids obtained from nulliparous mice, a significant reduction in *Pgr* expression was observed in E2 + HA condition compared to in E2 alone (*p*-value < 0.05; Figure 2C). No such difference was observed in organoids obtained from multiparous mice, corroborating the existence of a memory of pregnancy in mammary epithelial cells. Furthermore, the expression of *Esr2* was 6.9 ± 6.9-fold higher in organoids obtained from multiparous mice than in organoids obtained from nulliparous mice, independently from the culture conditions (all *p*-values < 0.05; Figure 2D). As both c-*Myc* and c-*Jun* are notably downstream targets of the canonical and non-canonical Wnt signaling pathways, respectively, which are both down-regulated by *Sfrp1*, as well as two oncogenes involved in breast cancer progression, we questioned if their expression could also be modulated by parity status, E2, or HA ex vivo, as *Sfrp1* was [31,32,33,34]. No difference in c-*Myc* expression was observed, either between nulliparous and multiparous mice, or according to culture conditions (Figure 2E). On the other hand, a significant decrease in c-*Jun* expression was observed in organoids obtained from multiparous mice, grown in E2 + HA, compared to both its control grown in the EtOH condition (1.7-fold ± 0.3), and the organoids from nulliparous mice grown in the E2 + HA condition (1.6-fold ± 0.6; all *p*-values < 0.05; Figure 2F). Taken together, these results support the hypothesis stipulating that the regulation of both Wnt and estrogen pathways in murine mammary epithelial cells are dependent from parity status, but also from the presence of HA microcalcifications.

In order to further investigate this hypothesis, we also compared the phenotype of these organoids. The total number of organoids over time was neither different between nulliparous and multiparous groups, nor between growth conditions. After 15 days of growth, the average numbers of organoids obtained in EtOH and EtOH + HA conditions, respectively, were 437.7 ± 87.3 and 384.3 ± 57.6 in the nulliparous group, while they were 406.7 ± 117.2 and 342.3 ± 93.0 in the multiparous group. In E2 and E2 + HA conditions, respectively, 463.7 ± 151.6 and 383.7 ± 173.1 organoids were counted in the nulliparous group, while 378.3 ± 117.3 and 332.3 ± 64.9 organoids were counted in the multiparous group. No significant difference in average organoid size between nulliparous and multiparous groups was observed (Figure 2G,H). However, organoids obtained from nulliparous mice in E2 condition (162.2 ± 26.2 μm) and in E2 + HA condition (168.3 ± 23.8 μm) were bigger than organoids obtained from nulliparous mice in EtOH conditions (131.5 ± 5.9 μm; all *p*-values < 0.05). Such a difference was not observed in organoids obtained from multiparous mice, corroborating the hypothesis that nulliparous and multiparous mammary glands do not respond to E2 stimulation in the same way (Figure 2G). 

Moreover, the difference in E2 response between nulliparous and multiparous groups was confirmed by the organoid phenotype analyses (Figure 2I–Q). After 15 days of growth in EtOH condition, 30.5 ± 1.2% of total organoids obtained from nulliparous mice were luminal, while 60.1 ± 4.7% were opaque and 9.4 ± 3.8% were lobular. On the other hand, in E2 condition, 88.2 ± 5.1% of total organoids obtained from nulliparous mice were luminal, only 9.3 ± 5.1% were opaque, and 2.5 ± 1.3% were lobular (Figure 2J,M,P; all *p*-values < 0.05). Similar results were obtained in E2 + HA condition compared to EtOH condition, without any difference between E2 and E2 + HA conditions. No difference between EtOH + HA and EtOH conditions was observed in organoids obtained from nulliparous mice. These results suggest that E2 promotes luminal organization of the mammary epithelial cells obtained from nulliparous mice.

On the other hand, E2 treatment had no statistically significant impact on the phenotype of organoids from multiparous mice. Indeed, after 15 days of growth in EtOH condition, 32.0 ± 9.4% of total organoids were luminal, 60.9 ± 8.6% were opaque, and 8.6 ± 3.4% were lobular, while in E2 condition, 46.8 ± 13.4% of total organoids were luminal, 49.5 ± 13.4% were opaque, and 3.7 ± 1.6% were lobular. No difference was observed between organoids obtained from multiparous mice grown in the presence of HA + EtOH compared to EtOH alone. However, after 15 days of growth, 56.9 ± 5.7% of total organoids obtained from multiparous mice cultivated in E2 + HA condition were luminal, 38.8 ± 4.0% were opaque, and 4.3 ± 1.7% were lobular, while in EtOH condition, only 30 ± 1.2% were luminal (*p*-value < 0.05; Figure 2K), 60.1 ± 4.7% were opaque (*p*-value < 0.05, Figure 2N), and 9.4 ± 3.8% were lobular (*p*-value = ns; Figure 2Q). These results suggest that mammary epithelial cells from parous mice respond weakly to E2 stimulation, while epithelial cells from nulliparous mice exhibit a stronger response. However, an additive effect from E2 + HA was observed in promoting the luminal phenotype in organoids obtained from multiparous mice only. Taken together, these results support the existence of a memory of pregnancy in murine mammary epithelial cells, which reduces their sensitivity to E2. The expression of *Sfrp1* in murine organoids does not seem to be impacted by parity status, while it was decreased by E2 treatment. Altogether, these results confirm what was described in women in Figure 1C; in other words, that *Sfrp1* expression is positively correlated with *Esr1* and *Pgr* expression in non-tumoral mammary epithelial cells and is associated with the breast lobulo-alveolar phenotype.

### 3.3. The Role of SFRP1 in Early Breast Carcinogenesis

#### 3.3.1. Non-Tumoral Breast Epithelial Cell

In order to verify if the *SFRP1* expression pattern across breast cell lines was similar to what we observed in breast cancer tissues from different molecular subtypes (Figure 1B), we used the publicly available UCSC Xena online tool (https://xenabrowser.net/ (accessed on 18 September 2021)). We saw that, indeed, breast cancer cell lines from luminal subtypes express less *SFRP1* than those from basal subtypes, and that normal breast cell lines express a higher *SFRP1* level than invasive breast cancer cell lines (Appendix A). Subsequently, in order to investigate the potential causal role of *SFRP1* decrease in expression in early breast carcinogenesis, we have knocked it down in an MCF10A non-tumoral cell line, considered as normal epithelial cells, by using two shRNAs (sh62169 and sh62170). qPCR analyses confirmed the efficiency of the two shRNAs compared to shScramble condition. In the MCF10A cell line, sh62169 induced a 57.3 ± 26.5% decrease in *SFRP1* expression, while sh62170 induced a 45.5 ± 26.6% decrease in *SFRP1* expression compared to shScramble in EtOH condition (all *p*-values < 0.05; Figure 3A). The expression of *SFRP1* was, nevertheless, not impacted by E2 treatments. The decrease in *SFRP1* expression induced a 2.4 ± 1.5-fold increase in *ESR1* expression in MCF10A sh62169 + EtOH and a 2.0 ± 1.2-fold increase in *ESR1* expression in MCF10A sh62170 + EtOH compared to MCF10A shScramble (all *p*-values < 0.06; Figure 3B). On the other hand, the decrease in *SFRP1* in expression had no impact on *ESR2* expression (Figure 3C). As *STAT3* is a well-known gene involved in both the initiation of breast lobular involution and early breast carcinogenesis, we questioned if a decrease in *SFRP1* expression could induce an increase in *STAT3* expression in non-tumoral breast cells [32,35,36,37,38]. Indeed, MCF10A sh62169 + EtOH showed a 1.3 ± 0.4-fold increase in *STAT3* expression compared to MCF10A shScramble + EtOH (*p*-value < 0.05; Figure 3D). In addition, MCF10A sh62169 + EtOH showed a 1.7 ± 0.6-fold decrease in *c-MYC* expression compared to MCF10A shScramble + EtOH (*p*-value < 0.05; Figure 3E). This was not observed in MCF10A sh62170 + EtOH, probably due to the low shRNA efficiency. The decrease in *SFRP1* expression did not change *c-JUN* expression in the MCF10A cell line (Figure 3F). Taken together, these results support the potential causal role of the decrease in *SFRP1* expression in the increase in *ESR1* expression in mammary epithelial cells.

To further assess if the decrease in *SFRP1* expression could be associated with the acquisition of tumoral properties, such as an increase in proliferation and migration abilities, we compared the phenotypes of the MCF10A cells following *SFRP1* knockdown. We observed a 14.0 ± 27.8% (*p*-value = ns) increase in MCF10A sh62169 + EtOH cell viability and a 26.9 ± 21.1% (*p*-value < 0.05) increase in MCF10A sh62170 + EtOH viability compared to MCF10A shScramble + EtOH after 72 h of growth. Interestingly, treatment with E2 induced a 47.7 ± 9.0% increase in MCF10A shScramble cell viability, a 62.0 ± 29.0% increase in MCF10A sh62169 cell viability, and an 84.0 ± 21.0% increase in MCF10A sh62170 cell viability compared to MCF10A shScramble + EtOH (Figure 3G, all *p*-values < 0.05), suggesting an additive effect of *SFRP1* knockdown and E2 treatment on promoting non-tumoral cell line MCF10A viability.

Spheroid assays confirmed that the decrease in *SFRP1* expression increases cell viability and proliferation (Figure 3H). Spheroids obtained from MCF10A sh62169 were17.5 ± 21.2% and 29.0 ± 4.3% bigger after 3- and 4-day growth, respectively, in EtOH condition compared to day 1 (all *p*-values < 0.05), while spheroids obtained from MCF10A shScramble did not grow (Figure 3H). Similarly, spheroids obtained from MCF10A sh62170 were 16.5 ± 3.2% bigger at day 2 and 9.0 ± 4.4% bigger at day 3 compared to day 1 (all *p*-values < 0.05) while spheroids obtained from MCF10A shScramble did not grow (Figure 3H). Surprisingly, treatment with E2 induced a 15.4 ± 5.6% decrease in MCF10A shScramble spheroid growth compared to MCF10A shScramble EtOH (*p*-value < 0.05) and an 18.6 ± 14.8% decrease in MCF10A sh62169 spheroid growth compared to MCF10A sh62169 EtOH (all *p*-values < 0.05). Combined with results presented in Figure 3G, these results suggest that E2 treatment could induce a compaction of the cells, hence a reduction in the spheroid size.

In addition, treatment with E2 induced a 3.2 ± 1.0-fold increase in branching in MCF10A shScramble compared to EtOH condition after 2 days of growth. MCF10A sh62170 had 3.8 ± 1.3 and 2.3 ± 0.8 times more ramifications than MCF10A shScramble after 2- and 3-day growth, respectively, in EtOH condition, validating that lack of *SFRP1* was associated with mammary branching (all *p*-values < 0.05; Figure 3I). Furthermore, compared to MCF10A shScramble + E2, MCF10A sh62170 + E2 had a 5.7 ± 2.9-fold increase in ramifications after 2 days of growth (*p*-value < 0.05), confirming the additive effects of both *SFRP1* knockdown and E2 treatment on the non-tumoral MCF10A breast cell line. MCF10A sh62170 + EtOH demonstrated a 29 ± 8.3% increase in cell migration after 24h compared to MCF10A shScramble + EtOH (*p*-value < 0.05, Figure 3J). On the other hand, in E2 condition, both MCF10A sh62169 and sh62170 reduced the cell migratory ability (40 ± 32% and 50 ± 13%, respectively; all *p*-values < 0.05), compared to shScramble, after 12 h of migration (Figure 3K). Altogether, these experiments support the potential causal role of the decrease in *SFRP1* expression in tumoral phenotype acquisition, but also in mammary gland branching in vitro.

#### 3.3.2. Atypical Ductal Hyperplasia

Similar experiments were performed on the MCF10AT1 cell line, which mimics ADH, in order to better understand when the “switch” between pre-tumoral and tumoral lesion occurs (Figure 4). In the MCF10AT1 cell line, sh62169 induced a 72.1 ± 4.0% decrease in *SFRP1* expression, while sh62170 induced a 62.6 ± 4.8% decrease in *SFRP1* expression compared to shScramble in EtOH condition (all *p*-values < 0.05; Figure 4A). Treatment with E2 had no impact on *SFRP1* expression, in MCF10AT1 shScramble or in MCF10AT1 shRNAs (Figure 4A). On the other hand, the decrease in *SFRP1* expression in MCF10AT1 sh62169 + EtOH and in MCF10AT1 sh62170 + EtOH induced a 2.9 ± 0.7-fold and a 2.9 ± 0.4-fold increase in *ESR1* expression, respectively, compared to MCF10AT1 shScramble + EtOH (all *p*-values < 0.05; Figure 4B). Treatment with E2 had no impact on *ESR1* expression, in MCF10AT1 shScramble or in MCF10AT1 shRNAs. The decrease in *SFRP1* expression also induced a 1.7 ± 0.4-fold increase in *ESR2* expression in MCF10AT1 sh62169 + EtOH compared to shScramble + EtOH (*p*-value < 0.05), and a 1.6 ± 1.1-fold increase in *ESR2* in MCF10AT1 sh62170 + EtOH compared to shScramble + EtOH (*p*-value = ns; Figure 4C). The decrease in *SFRP1* expression in MCF10AT1 sh62169 + EtOH and MCF10AT1 sh62170 + EtOH induced a 1.2 ± 0.03-fold and a 1.2 ± 0.3-fold decrease, respectively, in *STAT3* expression, compared to MCF10AT1 scramble + EtOH (all *p*-values < 0.05; Figure 4D). The decrease in *SFRP1* expression in MCF10AT1 sh62169 + EtOH and MCF10AT1 sh62170 + EtOH induced a 1.3 ± 0.3-fold (*p*-value = ns) and a 2.0 ± 0.9-fold (*p*-value < 0.05) decrease, respectively, in *c-MYC* expression, compared to MCF10AT1 scramble + EtOH (Figure 4E). Finally, treatment with E2 induced a 6.0 ± 5.0-fold increase in *c-JUN* expression in MCF10AT1 shScramble, while it had no significant impact on it in MCF10AT1 shRNAs (*p*-value < 0.05; Figure 4F).

In order to confirm what we observed previously in the non-tumoral MCF10A cell line, i.e., that the decrease in *SFRP1* induced an increase in epithelial cell tumoral properties, we have examined the proliferation and migratory abilities of the pre-tumoral MCF10AT1 cell line following the decrease in *SFRP1* expression. In contrast to MCF10A, treatment with E2 reduced MCF10AT1 shScramble viability by 10.5 ± 9.1% compared to MCF10AT1 + EtOH (*p*-value < 0.05; Figure 4G). Similar results were observed between MCF10AT1 sh62169 + E2 and MCF10AT1 sh62170 + E2 compared to their respective EtOH conditions; they were, nevertheless, statistically insignificant (Figure 4G). Surprisingly, while MCF10AT1 sh62169 + EtOH showed a 10.8 ± 7.0% increase in cell viability compared to MCF10AT1 shScramble + EtOH, MCF10AT1 sh62170 showed a 9.0 ± 5.9% decrease in cell viability compared to MCF10AT1 shScramble + EtOH (all *p*-values < 0.05; Figure 4G). The same tendency was observed in 3D culture; however, results were not statistically significant (Figure 4H). No difference in cell migration abilities were observed considering *SFRP1* expression in the MCF10AT1 cell line in EtOH condition (Figure 4I). However, in E2 condition, the decrease in *SFRP1* expression in MCF10AT1 induced an increase in the migratory ability of the cells. More precisely, MCF10AT1 sh62169 + E2 showed a 42.6 ± 19.5% and 25.6 ± 11.9% increase in wound healing after 24h and 30h of migration, respectively, compared to MCF10AT1 shScramble + E2 (all *p*-values < 0.05). Similarly, MCF10AT1 sh62170 + E2 showed a 17.3 ± 7.4%, a 44.0 ± 2.1%, and a 25.8 ± 11.7% increase in wound healing after 12 h, 24 h, and 30 h of migration, respectively, compared to MCF10AT1 shScramble + E2 (all *p*-values < 0.05; Figure 4J). Altogether, these results support the potential causal role of the lack of *SFRP1* in breast carcinogenesis. In addition, the reduction in *SFRP1* expression in the MCF10AT1 cell line seems to be in favor of non-canonical Wnt signaling pathway (Figure 4E,F).

### 3.4. Increase in SFRP1 Expression in Pre-Invasive Cell Line

A very low expression of SFRP1 was detected in the MCF10DCIS cell line, which mimics a ductal carcinoma in situ. Thus, in order to better understand the impact of *SFRP1* on breast cancer progression, we decided to overexpress it in this cell line, investigating the potential of *SFRP1* to decrease pre-invasive breast cancer aggressiveness in vitro. In MCF10DCIS OxSFRP1 + EtOH, *SFRP1* expression was increased 27.4 ± 14.9-fold compared to MCF10DCIS empty pLX_TRC304 + EtOH, and by 50.0 ± 6.9-fold in E2 conditions (all *p*-values < 0.05, Figure 5A). Interestingly, the overexpression of *SFRP1* in MCF10DCIS + EtOH induced a 1.5 ± 0.3-fold increase in *ESR1* expression compared to MCF10DCIS empty pLX_TRC304 + EtOH (*p*-value < 0.05; Figure 5B). As a reminder, the decrease in *SFRP1* expression in non-tumoral (MCF10A) and pre-tumoral (MCF10AT1) breast lesions was inversely correlated with *ESR1* expression (Figure 3B and Figure 4B), highlighting the existence of a differential role of *SFRP1* with regards to breast cancer stage and molecular subtype. Treatment with E2 had no impact on *ESR1* expression, in MCF10DCIS OxSFRP1 or in MCF10DCIS empty pLX_TRC304. On the other hand, the increase in *SFRP1* expression in MCF10DCIS induced a 1.7 ± 0.9-fold decrease and a 1.8 ± 0.82-fold decrease in *ESR2* expression compared to MCF10DCIS empty pLX_TRC304 in EtOH and E2 conditions, respectively (all *p*-values < 0.05; Figure 5C). *SFRP1* overexpression had no impact on either *STAT3* (Figure 5D) expression or in *c-MYC* (Figure 5E) and *c-JUN* (Figure 5F) expression in MCF10DCIS, regardless of the culture condition. Altogether, these results suggest that in the MCF10DCIS cell line, *SFRP1* remains positively correlated with *ESR1* expression and negatively correlated with *ESR2* expression, as in non-tumoral breast tissue (Figure 1C).

In order to assess the potential of *SFRP1* to act as a tumor suppressor in pre-invasive lesions, we also examined the phenotype of the MCF10DCIS cell line following the increase in *SFRP1* expression. The increase in *SFRP1* expression in ductal carcinoma in situ in the MCF10DCIS cell line modified neither cell viability (Figure 5G) nor spheroid growth (Figure 5H), regardless of the culture condition. We observed a slight decrease in MCF10DCIS OxSFRP1 + EtOH spheroid growth compared to MCF10DCIS empty pLX_TRC304 + EtOH after 2 days of growth (*p*-value < 0.05). However, this difference disappeared after 3 days of growth. On the other hand, *SFRP1* overexpression was responsible for a decrease in MCF10DCIS cell line migratory abilities in EtOH condition (Figure 5I), but not in E2 condition (Figure 5J). More precisely, in EtOH condition, *SFRP1* overexpression induced a 41.4 ± 14.4% and a 31.4 ± 11.5% decrease in wound healing after 24h and 36h of migration, respectively, compared to MCF10DCIS empty pLX_TRC304 (all *p*-values < 0.05). Inversely, in E2 condition, the overexpression of *SFRP1* induced a 12.9 ± 8.4% increase in wound healing after 36 h of migration, compared to MCF10DCIS empty pLX_TRC304 (*p* = value < 0.05). Altogether, these results suggest that *SFRP1* reduced the migratory abilities of the MCF10DCIS cell line in the absence of E2 only. On the other hand, these results also raise the question of the potential oncogenic role of *SFRP1* in pre-invasive lesions in the presence of E2.

### 3.5. Increase in SFRP1 Expression in Triple-Negative Invasive Cell Line

In order to further understand the impact of *SFRP1* on breast cancer progression, we then overexpressed it in the basal-like invasive ductal carcinoma MCF10CA1a cell line. Like in MCF10DCIS, we detected very low *SFRP1* expression in MCF10CA1a. Furthermore, as we previously observed that *SFRP1* overexpression in MCF10DCIS induced an increase in *ESR1* expression (Figure 5B), we hypothesized that it would also be the case in the MCF10CA1a cell line. Consequently, aiming to detect a possible change in E2 responsiveness following *SFRP1* overexpression, we tested the effects of E2 treatment on MCF10CA1a tumorigenic properties, even if this cell line does not express ERα basically.

In MCF10CA1a OxSFRP1 + EtOH, *SFRP1* expression increased 207 ± 42-fold, compared to MCF10CA1a empty pLX_TRC304 + EtOH, and 290.3 ± 26.4-fold in E2 conditions (all *p*-values < 0.05, Figure 6A). On the other hand, while *SFRP1* overexpression had no impact on *ESR1* expression in MCF10CA1a, treatment with E2 significantly reduced *ESR1* expression in MCF10CA1a OxSFRP1, compared to both MCF10CA1a empty pLX_TRC304 + EtOH (1.5 ± 0.2-fold) and MCF10CA1a OxSFRP1 + EtOH (1.4 ± 0.1-fold (all *p*-values < 0.05; Figure 6B), suggesting that *SFRP1* expression sensitizes MCF10CA1a to E2. Similarly, the increase in *SFRP1* expression in the MCF10CA1a cell line did not modify *ESR2* expression in EtOH condition, while it exacerbated the E2-dependent decrease in *ESR2* expression (*p*-values < 0.05; Figure 6C). Indeed, treatment with E2 induced a 1.8 ± 0.4 decrease in *ESR2* expression in MCF10CA1a empty pLX_TRC304 (*p*-value < 0.05; Figure 6C). In addition, in E2 condition, *ESR2* expression was 1.7 ± 0.5-fold lower in MCF10CA1a OxSFRP1 compared to MCF10CA1a empty pLX_TRC304 (*p*-value < 0.05; Figure 6C). On the other hand, *SFRP1* overexpression in MCF10CA1a induced a 1.6 ± 0.2-fold increase in *STAT3* expression compared to MCF10CA1a empty pLX_TRC304 in EtOH condition (*p*-value < 0.05; Figure 6D). However, *STAT3* expression was brought down to a level similar to that of MCF10CA1a empty pLX_TCR304 + EtOH after treatment with E2. Furthermore, in the MCF10CA1a cell line, the increase in *SFRP1* expression decreased both canonical and non-canonical Wnt signaling pathways. Indeed, it decreased both *c-MYC* (1.7 ± 0.5-fold; *p*-value < 0.05 Figure 6E) and *c-JUN* (1.6 ± 0.6-fold; *p*-value < 0.05; Figure 6F) expression in MCF10CA1a OxSFRP1 compared to MCF10CA1a empty pLX_TCR304 in EtOH condition. This decrease was exacerbated in E2 condition (Figure 6E,F).

Interestingly, neither treatment with E2 nor *SFRP1* overexpression modified MCF10CA1a viability compared to MCF10CA1a empty pLX_TRC304 + EtOH (Figure 6G). However, as expected according to the gene expression analyses described previously, *SFRP1* overexpression increased MCF10CA1a sensitivity to E2. Indeed, MCF10CA1a OxSFRP1 + E2 demonstrated a 14.2 ± 2.3% decrease in cell viability compared to MCF10CA1a empty pLX_TRC304 + EtOH and a 16.8 ± 4.3% decrease in cell viability compared to MCF10CA1a OxSFRP1 + EtOH (all *p*-values < 0.05; Figure 6G). This was confirmed by spheroid assays (Figure 6H), as after only two days of growth, MCF10CA1a empty pLX_TRC304 + E2 demonstrated a 10.2 ± 5.6% decrease in spheroids growth, MCF10CA1a OxSFRP1 + EtOH demonstrated a 12.0 ± 5.3% decrease in spheroid growth and MCF10CA1a OxSFRP1 + E2 demonstrated a 15.8 ± 8.5% decrease in spheroid growth compared to MCF10CA1a empty pLX_TRC304 + EtOH (all *p*-values < 0.05). After 5 days of growth, MCF10CA1a pLX_TRC304 + E2 demonstrated a 12.3 ± 13.2% decrease in spheroid growth, while MCF10CA1a OxSFRP1 + EtOH demonstrated a 23.3 ± 0.5% decrease in spheroid growth and MCF10CA1a OxSFRP1 + E2 demonstrated a 26.9 ± 11.6% decrease in spheroid growth compared to MCF10CA1a pLX_TRC304 + EtOH (all *p*-values < 0.05). In addition, MCF10CA1a OxSFRP1 + EtOH showed an 8.2 ± 10.3% decrease in spheroid growth, while MCF10CA1a OxSFRP1 + E2 showed a 12.1 ± 4.4% decrease in spheroid growth compared to MCF10CA1a pLX_TRC304 + E2 (all *p*-values < 0.05). In addition to the reduction in MCF10CA1a viability, *SFRP1* overexpression also reduced cell migratory abilities, in both EtOH (Figure 6I) and E2 (Figure 6J) conditions. More precisely, in EtOH condition, SFRP1 overexpression induced a 24.4 ± 16.1% decrease and a 38.1 ± 16.1% decrease in MCF10CA1a wound healing after 15 h and 24 h, respectively, compared to MCF10CA1a empty pLX_TRC304 (all *p*-values < 0.05). In E2 condition, *SFRP1* overexpression induced a 17.0 ± 2.6% decrease and an 18.8 ± 8.1% decrease in MCF10CA1a wound healing after 24 h and 36 h of migration, respectively, compared to MCF10CA1a empty pLX_TRC304 (all *p*-values < 0.05). Taken together, these results confirm the potential of SFRP1 to act as a tumor suppressor in basal-like triple-negative breast cancer, notably by reducing both canonical (Figure 6E) and non-canonical (Figure 6F) Wnt signaling pathways.

### 3.6. Increase in SFRP1 Expression in Luminal A Invasive Cell Line

In order to further assess if the increase in *SFRP1* expression could potentially decrease luminal A breast cancer progression, we overexpressed *SFRP1* in the MCF7 cell line (Figure 7). Indeed, as was observed in luminal A invasive breast tumoral tissue in women, the MCF7 cell line expresses a very low SFRP1 level [20]. Firstly, we observed a significant increase in *SFRP1* expression following the lentiviral transduction (*p*-value < 0.05; Figure 7A). Treatment with E2 had no impact on *SFRP1* expression in the MCF7 empty pLX_TRC304, or MCF7 OxSFRP1 cell lines (Figure 7A). While not statistically significant, the increase in *SFRP1* expression in the MCF7 cell line induced a 1.2 ± 0.4-fold decrease in *ESR1* expression compared to MCF7 empty pLX_TRC304 in EtOH condition (Figure 7B). Treatment with E2 induced a 1.2 ± 0.1-fold increase in ESR1 expression in MCF7 OxSFRP1 compared to MCF7 OxSFRP1 grown in EtOH condition (*p*-value < 0.05; Figure 7B). In addition, the increase in *SFRP1* expression in the MCF7 cell line induced a 1.2 ± 0.2-fold increase in *ESR2* expression (*p*-value < 0.05; Figure 7C) but it did not modify *STAT3* expression (Figure 7D). The increase in *SFRP1* expression in the MCF7 cell line also induced a 1.3 ± 0.2-fold increase in *c-MYC* expression (*p*-value < 0.05; Figure 7E), while it did not change c-JUN expression (Figure 7F) compared to MCF7 empty pLX_TRC304 + EtOH. These results suggest that the increase in *SFRP1* expression in the luminal A cell line reduces *ESR1* expression (Figure 7B), while it increases *ESR2* expression (Figure 7C), and favors the canonical Wnt signaling pathway (Figure 7E).

In order to further assess the potential role of *SFRP1* in reducing luminal A breast cancer aggressiveness, we compared the proliferative and migratory abilities of the MCF7 cell line according to *SFRP1* modulation of expression. The increase in *SFRP1* expression induced a 20.0 ± 5.0% reduction in MCF7 viability compared to MCF7 empty pLX_TRC304 (*p*-value < 0.05; Figure 7G). Treatment with E2 did not modify MCF7 viability compared to EtOH condition (Figure 7G). Unexpectedly, spheroid size was higher in MCF7 OxSFRP1, in both EtOH and E2 conditions compared to MCF7 empty pLX_TRC304 grown in EtOH condition after 3 days of growth (*p*-values < 0.05; Figure 7H). Spheroid size was also higher in MCF7 OxSFRP1 grown in E2 condition compared to MCF7 empty pLX_TRC304 grown in EtOH condition after 5 days of growth (*p*-values < 0.05; Figure 7H). This could notably be due to a reduction in spheroid density or an increase in cell size in the MCF7 OxSFRP1 cell line. These results suggest, nevertheless, that *SFRP1* could reduce luminal A breast cancer aggressiveness by reducing cell viability. In addition to proliferative ability, we investigated the ability of the cells to migrate by performing a wound-healing assay. In EtOH condition, *SFRP1* overexpression induced a 33.2 ± 10.6%, a 43.1 ± 13.2%, and a 46.2 ± 29.8% decrease in wound healing after 12 h, 24 h, and 36 h of migration, respectively, compared to MCF7 empty pLX_TRC304 (all *p*-values < 0.05, Figure 7I). Similarly, in E2 condition, *SFRP1* overexpression induced a 31.8 ± 9.5%, a 47.5 ± 4.5%, and a 58.3 ± 8.9% decrease in wound healing after 12h, 24h, and 36h of migration, respectively, compared to MCF7 empty pLX_TRC304 (all *p*-values < 0.05, Figure 7J).

Altogether, these results support the potential of *SFRP1* to decrease luminal A breast lesion aggressiveness by reducing *ESR1* expression (Figure 7B), while increasing *ESR2* expression (Figure 7C), but also by decreasing both tumoral cell viability (Figure 7G) and migratory abilities (Figure 7I,J).

### 3.7. Association between SFRP1 Expression and Breast Cancer Outcomes

To corroborate these results, we obtained in vitro in breast cancer cell lines and performed public dataset analyses in order to investigate the association between *SFRP1* expression and breast cancer patient outcomes. High *SFPR1* expression was significantly associated with lower recurrence-free survival (RFS; HR = 0.68; logrank *p* = < 0.0001; Figure 8A) and overall survival (OS; HR = 0.59; logrank *p* = < 0.01; Appendix A) in ERα-positive breast cancer patients in the KMplot.com gene expression public dataset [27]. Luminal A breast cancer patients with high *SFRP1* expression had a better RFS (HR = 0.58; logrank *p* = < 0.01; Figure 8B) and OS (HR = 0.73; logrank *p* = < 0.001; Appendix A) than patients with low *SFRP1* expression. Luminal B breast cancer patients with high SFRP1 expression had a better prognosis in term of RFS (HR = 0.74; logrank *p* = < 0.01; Figure 8C) and OS (HR = 0.66; logrank *p* = < 0.05; Appendix A) than patients with low SFRP1 expression. Interestingly, in ERα-negative breast cancer patients, high *SFRP1* expression was associated with lower RFS (HR = 0.76; logrank *p* = < 0.01; Figure 8D) but not with OS (HR = 0.80; OS logrank *p* = 0.20; Appendix A). No difference was observed in RFS and OS rates between high SFRP1 expression and low *SFRP1* expression groups in HER2-positive (Figure 8E and Appendix A) and basal breast cancer patients (Figure 8F and Appendix A). Consequently, the PRECOG [29] meta-Z-score for *SFRP1* expression in breast cancer was −0.2, suggesting an absence of association between this gene’s expression and the patient’s outcome. However, as described previously, *SFRP1* expression was associated with luminal breast cancer outcomes specifically, expressing ERα and/or PR, corroborating the existence of a crosstalk between the Wnt signaling pathway and the ERα signaling pathway during breast carcinogenesis. ROCplot analyses also confirmed that patients treated with adjuvant tamoxifen who relapsed in the 5 years following diagnosis (n = 127) had a lower *SFRP1* expression than patients who did not relapse (n = 759; Mann–Whitney *p*-value = 0.0012; Figure 8G). In addition, patients with a complete pathological response to chemotherapy (n = 532) also had a higher *SFRP1* expression than those who did not respond (n = 1100; Mann–Whitney *p*-value = 3.4 × 10^−7^ [39]). After stratification for molecular subtypes, this association was only conserved in patients with luminal A breast cancer (responders n = 134; non-responders n = 341; Mann–Whitney *p*-value = 2.9 × 10^−8^; Figure 8H), as well as in patients with TNBC (responders n = 196; non-responders n = 277; Mann–Whitney *p*-value = 0.046; Figure 8I), but not in luminal B and HER2 patients [39]. No information regarding the basal-like specific subgroup was available in this dataset.

## 4. Discussion

In the present manuscript, we first corroborated previous studies from our group and others, in which a decrease in *SFRP1* expression was observed in the continuum of breast cancer risk and in breast cancer, compared to non-tumoral tissue, by analyzing public datasets [20,40,41,42,43,44,45,46,47,48]. Public dataset analyses have also confirmed that the decrease in *SFRP1* expression among tumoral tissue compared to non-tumoral tissue could be mediated by its promoter, hypermethylation, in luminal and HER2 invasive carcinomas [40,42,46].

Our results also supported those from another study performed by our group, in which we observed a differential regulation of *SFRP1* expression, as well as a different mammary gland phenotype, according to parity and microcalcification status [23]. Indeed, we observed that organoids from nulliparous mice expressed more Esr1 than those from multiparous mice, and that treatment with E2 reduced *Sfrp1* and *Esr1* expression in both groups. We also observed that the decrease in both *Sfrp1* and *Esr1* expression following E2 treatment was associated with a lobulo-alveolar phenotype in organoids obtained from nulliparous mice only. Interestingly, nulliparous women have a majority of type 1 lobules (>12 acini), which are more positive to ERα and PR than type 3 lobules (>80 acini), while parous women have more type 3 lobules, expressing less ERα and PR than type 1 lobules, which is concordant with the fact that organoids obtained from nulliparous mice expressed more *Esr1* than those obtained from multiparous mice [12,13,14,15,49,50]. In addition, virgin mice depleted of *Sfrp1* developed branched mammary glands, similar to pregnant mice in vivo, corroborating that the decrease in *Sfrp1* expression by E2 treatment in organoids obtained from nulliparous mice increased organoid luminal organization ex vivo [22].

In the present study, we also observed that organoids obtained from nulliparous mice developed luminal organization upon E2 treatment, while organoids obtained from multiparous mice seemed resistant to E2 treatment and became progressively opaque, while both demonstrated a decrease in *Sfrp1* expression. Interestingly, a study by dos Santos et al. reported the existence of an epigenetic memory of pregnancy in mice mammary glands [51]. Feigman et al. also showed that mammary epithelial cells from parous mice developed resistance to c-Myc downstream pathways compared to mammary epithelial cells from nulliparous mice [52]. If mammary epithelial cells became resistant to c-Myc after the first pregnancy, then modulation of *Sfrp1* expression by E2 treatment would not modify mammary gland phenotype, as we observed ex vivo in organoids obtained from multiparous mice. Moreover, Kanadys et al. observed that the duration of oral contraceptive use in women was associated with breast cancer risk, when the calculation of period of consumption was limited to the period preceding the first full-term pregnancy [53], corroborating the potential acquisition of an estrogen resistance following the first full-term pregnancy. The c-Myc resistance development after the first pregnancy in women remains underexplored and could be an important avenue in the comprehension of the multiparity protective effect against breast cancer development. As a limit of the presented ex vivo work, it remains important to highlight that some differences between conditions could have been missed, as right and left inguinal mammary glands have been sampled and pooled in our experiments. Indeed, mouse embryos develop five pairs of mammary glands with distinct sequential and positional molecular processes [54]. Few studies have postulated that in women, the higher incidence of left breast cancer compared to right breast could be due to these sequential and positional genetic programs [54,55,56,57,58,59,60,61,62]. Further experiments on mice mammary glands should take this evidence into account. In addition, we failed to validate ex vivo what we previously observed in women, i.e., that *SFRP1* was decreased in nulliparous compared to parous women following age-related lobular involution [23]. Here, nulliparous mice sampled were 235.1 ± 56.9 days old, while multiparous mice were 319.4 ± 48.8 days old. Such a difference in age could potentially explain the absence of difference in *Sfrp1* expression between both groups.

We also demonstrated that a decrease in *SFRP1* expression in both non-tumoral MCF10A and hyperplasic MCF10AT1 cell lines induced an increase in the tumorigenic properties of the cells in vitro. In addition, a decrease in *SFRP1* expression in non-tumoral and pre-tumoral cell lines also increased their sensitivity to E2 and the expression of *ESR1*. Interestingly, the public dataset analyses we performed also reported that *SFRP1* expression was positively correlated with tumoral biomarkers such as *ESR1*, *PGR,* and *HER2* in non-tumoral tissue, while it became negatively correlated with such genes in breast tumoral tissues in women, raising the potential causal role of *SFRP1* dysregulation in breast carcinogenesis initiation. However, as a limit of these analyses, it is important to keep in mind that these correlations were not adjusted for age or for parity status. That being said, combined with the results we obtained in vitro on both MCF10A and MCF10AT1 cell lines, it is tempting to speculate that the “switch” in *SFRP1* expression positive correlation with *ESR1* expression is already observable in MCF10A and MCF10AT1 cell lines, corroborating the hypothesis that *SFRP1* could be involved in early ERα-positive breast carcinogenesis. These results are also in concordance with the study of Gregory et al., in which the authors reported that breast hyperplasia samples expressed less SFRP1 than paired benign tissue, as well as that treatment of human breast explant cultures with E2 + recombinant SFRP1 decreased the response to E2 compared to treatment with E2 alone [48]. However, functional analyses of the crosstalk between SFRP1 and ERα at the protein level must be completed in order to reach a full conclusion. Indeed, the presented analyses have been completed at mRNA level only and do not confirm the existence of an ERα-dependent carcinogenesis induced by the decrease in *SFRP1* expression.

Subsequently, we confirmed the potential of SFRP1 to reduce luminal A aggressiveness by increasing its expression in the MCF7 cell line. Indeed, we observed a reduction in *ESR1* expression, and a decrease in cell viability and migratory abilities in MCF7 overexpressing *SFRP1* compared to the negative control. This was also in concordance with public dataset analyses, which demonstrated that in patients treated with tamoxifen adjuvant therapy, the 5-year relapse rate was lowest in patients expressing more SFRP1 than the median of expression, compared to patients who expressed lower *SFRP1* than the median. Once more, these analyses were not adjusted for age and parity status, mainly because this last variable remains rarely available in public datasets. Therefore, the knowledge regarding the molecular impact of parity on breast cell phenotype is not well understood. Nevertheless, a few studies showed that high parity (>3 children) has been associated with an increased breast cancer mortality compared to nulliparous women [63], and having four children or more was also associated with a worse invasive breast cancer outcome when compared to women with one child [64]. Considering that SFRP1 is associated with both breast cancer outcome and parity status, future survival analyses aiming to investigate the potential of SFRP1 to be used as a prognostic factor should be adjusted for parity status.

Finally, we showed that *SFRP1* overexpression in MCF10DCIS pre-invasive and MCF10CA1a invasive triple-negative breast cancer cell lines reduced their aggressiveness in vitro. However, we observed that *SFRP1* overexpression in MCF10DCIS increased *ESR1* expression, and then increased cell migration abilities in E2 conditions, while it decreased this expression in EtOH condition, suggesting that the role of *SFRP1* in ERα-negative ductal carcinoma in situ could be E2-dependent. On the other hand, *SFRP1* overexpression in the MCF10CA1a cell line did not modify *ESR1* expression in EtOH condition. However, in E2 condition, *SFRP1* expression was further increased, and *ESR1* expression further decreased, resulting in an exacerbation of the reduction in cell viability compared to *SFRP1* overexpression alone. In accordance with a previous work from Bernemann et al., we found that, in the public dataset, TNBC patients who had a complete pathological response to chemotherapy had a higher *SFRP1* expression than non-responder, suggesting that *SFRP1* could be a potential tumor suppressor in TNBC, as well as a good prognostic marker [43]. Nevertheless, complementary experiments on other TNBC subtypes should be performed, as TNBC remains a highly heterogenous disease.

## 5. Conclusions

To conclude, the presented results showed that *SFRP1* expression is positively correlated with *ESR1* in non-tumoral breast tissue, while it is negatively correlated with *ESR1* in invasive breast cancer. In addition, *Sfrp1* expression is reduced upon E2 treatment in murine mammary organoids, independently from the mouse’s parity status. They also confirmed the existence of the memory of pregnancy, responsible for a decrease in organoid sensitivity to E2. In addition, the decrease in *SFRP1* expression in non-tumoral cell lines induced an increase in mammary branching in vitro and an increase in tumorigenic properties, such as viability and migration, suggesting a potential causal role of the decrease in *SFRP1* expression in early breast carcinogenesis. On the other hand, the overexpression of *SFRP1* in pre-invasive and invasive breast cancer cell lines induced a reduction in cell viability and migratory abilities, suggesting that *SFRP1* could be a potential tool against breast cancer progression. Further analysis to better understand the functional crosstalk between SFRP1 and ERα should be performed in order to assess the causal role of SFRP1 in luminal carcinogenesis.

## Figures and Tables

**Figure 1 cancers-15-02251-f001:**
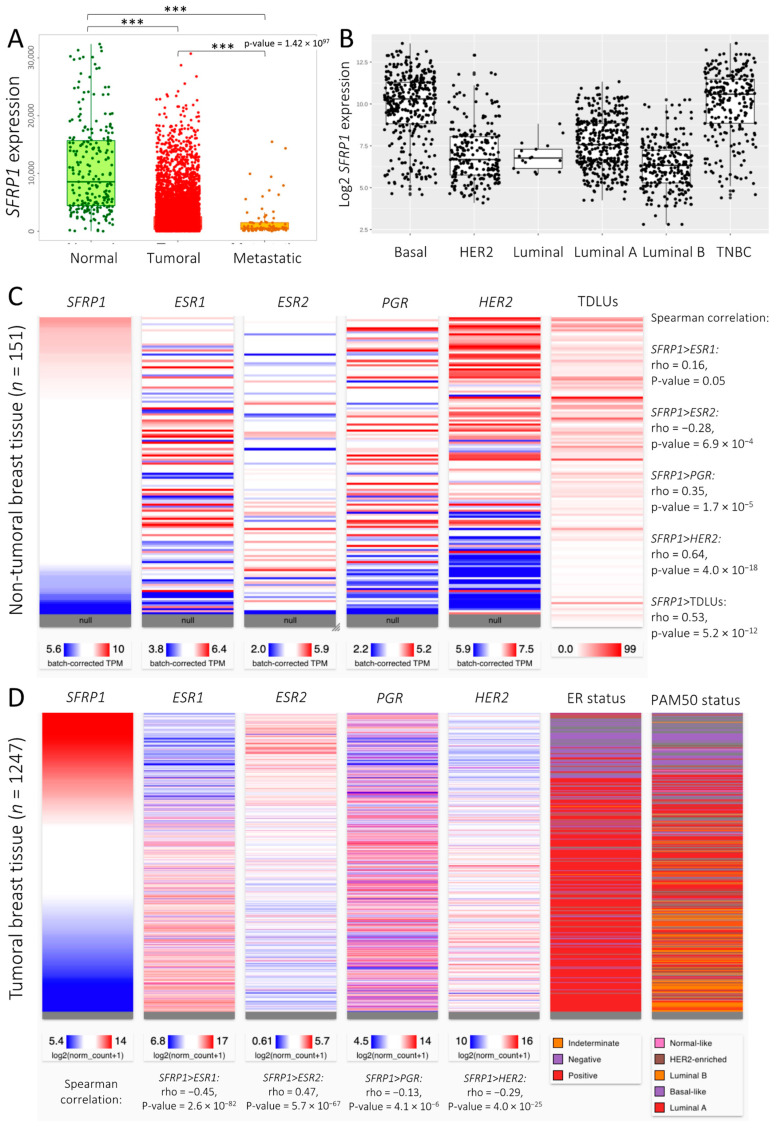
*SFRP1* expression pattern across breast cancer tissues. (**A**) *SFRP1* expression profile in tumoral (n = 7569; FC = 0.20) compared to non-tumoral (n = 242; Dunn’s test *p*-value < 0.0001) breast tissues and metastatic (n = 82; FC = 0.77) compared to tumoral (n = 7596; Dunn’s test *p*-value < 0.0001) breast tissues. (**A**) was obtained from http://tnmplot.com/platform (accessed on 15 April 2021). (**B**) *SFRP1* expression pattern across breast cancer molecular subtypes assessed by immunohistochemistry. (**B**) was obtained from http://gent2.appex.kr/gent2/ (accessed on 15 April 2021). (**C**) Co-expression profile of *SFRP1*, *ESR1*, *ESR2*, *PGR,* and *HER2* in non-tumoral breast tissue from Benz et al. database (n = 151). (**D**) Co-expression profile of *SFRP1*, *ESR1*, *ESR2*, *PGR,* and *HER2* in invasive ductal carcinoma, from TCGA database, classified by molecular subtypes following the PAM50 diagnostic method (n = 1247). (**C**,**D**) were drawn with UCSC Xena online tool (https://xenabrowser.net/ (accessed on 18 September 2021)). *** = *p*-value < 0.001.

**Figure 2 cancers-15-02251-f002:**
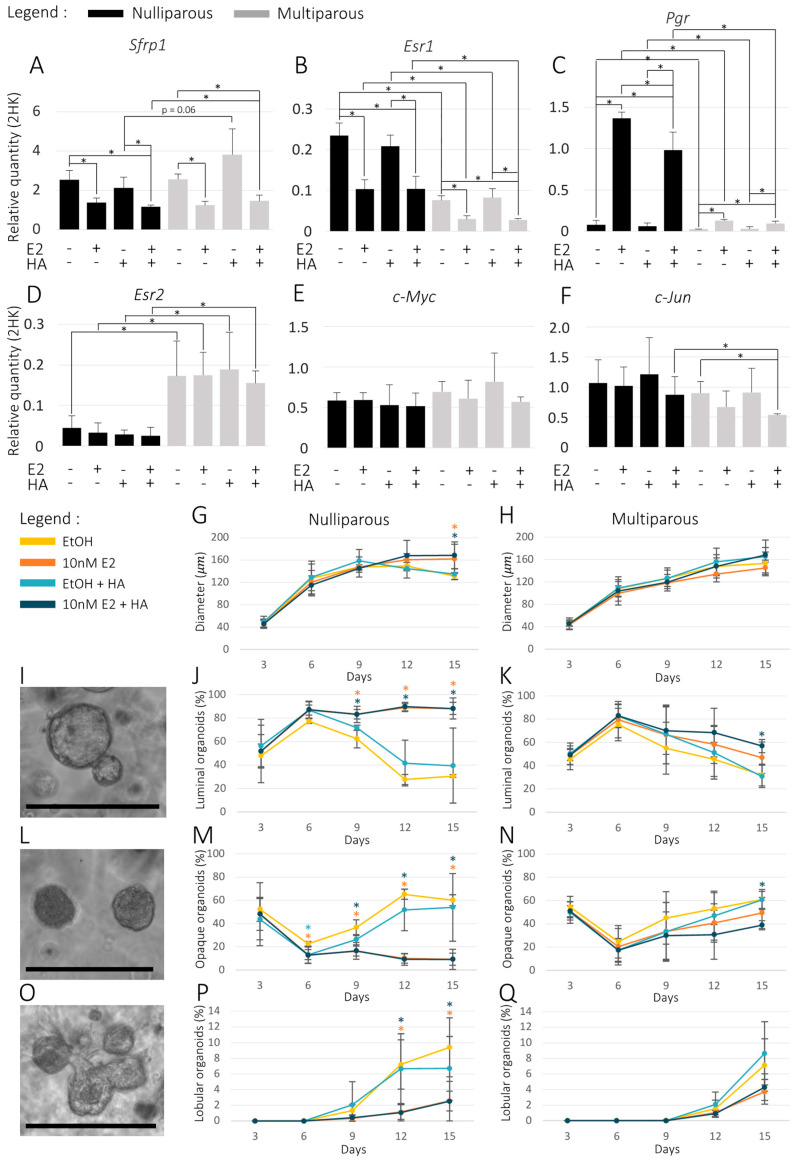
Characterization of phenotype and *Sfrp1* expression pattern in mammary gland organoids obtained from nulliparous and multiparous mice. *Sfrp1* (**A**), *Esr1* (**B**), *Pgr* (**C**), *Esr2* (**D**), *c-Myc* (**E**), and *c-Jun* (**F**) levels of expression among organoids obtained from both nulliparous and multiparous mice, in EtOH, EtOH + HA, E2, and E2 + HA conditions quantified by qPCR. Average organoid diameter obtained from (**G**) nulliparous and (**H**) multiparous mice. Percentage of total organoids with (**I**) luminal phenotype obtained from (**J**) nulliparous and (**K**) multiparous mice. Percentage of total organoids with (**L**) opaque phenotype obtained from (**M**) nulliparous and (**N**) multiparous mice. Percentage of total organoids with (**O**) lobular phenotype obtained from (**P**) nulliparous and (**Q**) multiparous mice. For (**I**,**L**,**O**), scale bars = 300 μm. All experiments were performed in technical triplicate and repeated three times independently. * = *p*-value < 0.05.

**Figure 3 cancers-15-02251-f003:**
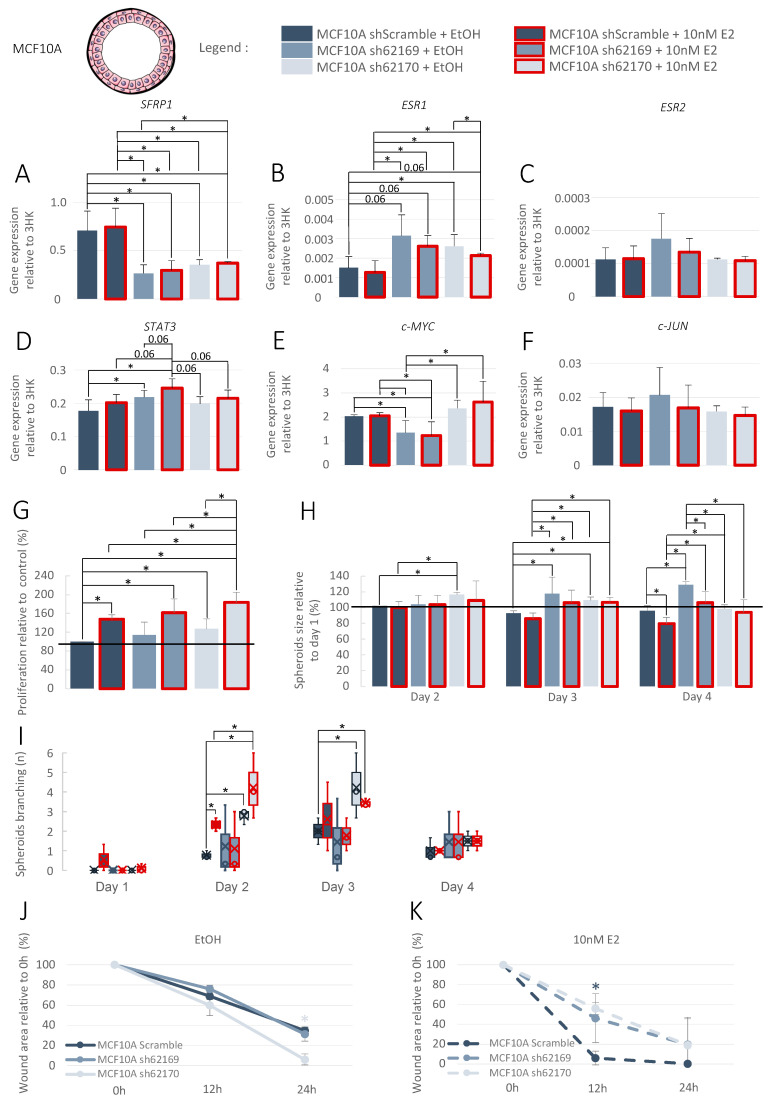
Decrease in *SFRP1* expression in non-tumoral MCF10A cell line. (**A**) *SFRP1*, (**B**) *ESR1*, (**C**) *ESR2*, (**D**) *STAT3*, (**E**) *c-MYC,* and (**F**) *c-JUN* expression relative to three housekeeping genes (HK) quantified by qPCR in MCF10A shRNAs and shScramble negative control, in EtOH and E2 conditions. (**G**) Viability assay comparing MCF10A shScramble negative control to MCF10A sh62169 and sh62170 targeting *SFRP1* in both EtOH and 10 nM E2 conditions. (**H**) Average spheroid size and (**I**) number of ramifications regarding *SFRP1* modulation of expression in MCF10A cell line, in both EtOH and 10 nM E2 conditions. (**J**,**K**) Wound-healing assay comparing MCF10A shScramble negative control to MCF10A sh62169 and sh62170 targeting *SFRP1* in (**J**) EtOH and (**K**) 10 nM E2 conditions. All experiments were performed in technical triplicate and repeated three times independently. * = *p*-value < 0.05.

**Figure 4 cancers-15-02251-f004:**
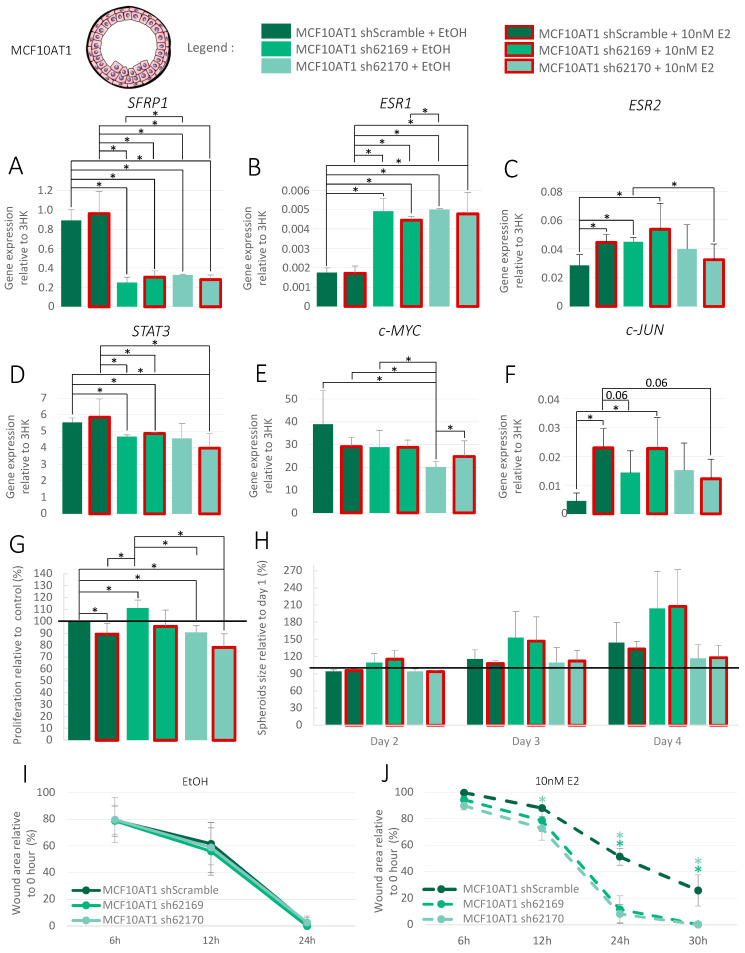
Decrease in *SFRP1* expression in pre-tumoral MCF10AT1 cell line. (**A**) *SFRP1*, (**B**) *ESR1*, (**C**) *ESR2*, (**D**) *STAT3*, (**E**) *c-MYC,* and (**F**) *c-JUN* expression relative to three housekeeping genes (HK) quantified by qPCR in MCF10AT1 shRNAs and shScramble negative control, in EtOH and E2 conditions. (**G**) Viability assay comparing MCF10AT1 shScramble negative control to MCF10AT1 sh62169 and sh62170 targeting *SFRP1* in both EtOH and 10 nM E2 conditions. (**H**) Average spheroid size regarding *SFRP1* modulation of expression in MCF10AT1 cell line, in both EtOH and 10 nM E2 conditions. (**I**,**J**) Wound-healing assay comparing MCF10AT1 shScramble negative control to MCF10AT1 sh62169 and sh62170 targeting *SFRP1*, in (**I**) EtOH and (**J**) 10 nM E2 conditions. All experiments were performed in technical triplicate and repeated three times independently. * = *p*-value < 0.05.

**Figure 5 cancers-15-02251-f005:**
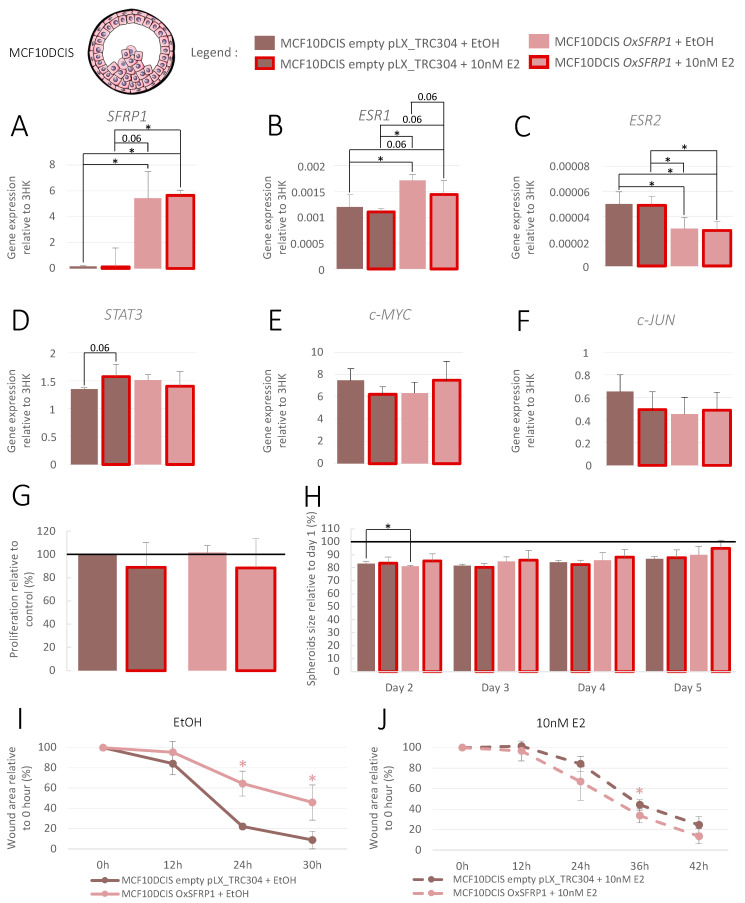
Increase in *SFRP1* expression in MCF10DCIS cell line. (**A**) *SFRP1*, (**B**) *ESR1*, (**C**) *ESR2*, (**D**) *STAT3*, (**E**) *c-MYC,* and (**F**) *c-JUN* expression relative to three housekeeping genes (HK) quantified by qPCR in MCF10DCIS OxSFRP1 and empty pLX_TRC304 negative control, in EtOH and E2 conditions. (**G**) Viability assay comparing MCF10DCIS empty pLX_TRC304 negative control to MCF10DCIS OxSFRP1 in both EtOH and E2 conditions. (**H**) Average spheroid size and regarding SFRP1 modulation of expression in MCF10DCIS cell line, in both EtOH and E2 conditions. Wound-healing assay comparing MCF10DCIS empty pLX_TRC304 negative control to MCF10DCIS OxSFRP1 in (**I**) EtOH and (**J**) 10 nM E2 conditions. All experiments were performed in technical triplicate and repeated three times independently. * = *p*-value < 0.05.

**Figure 6 cancers-15-02251-f006:**
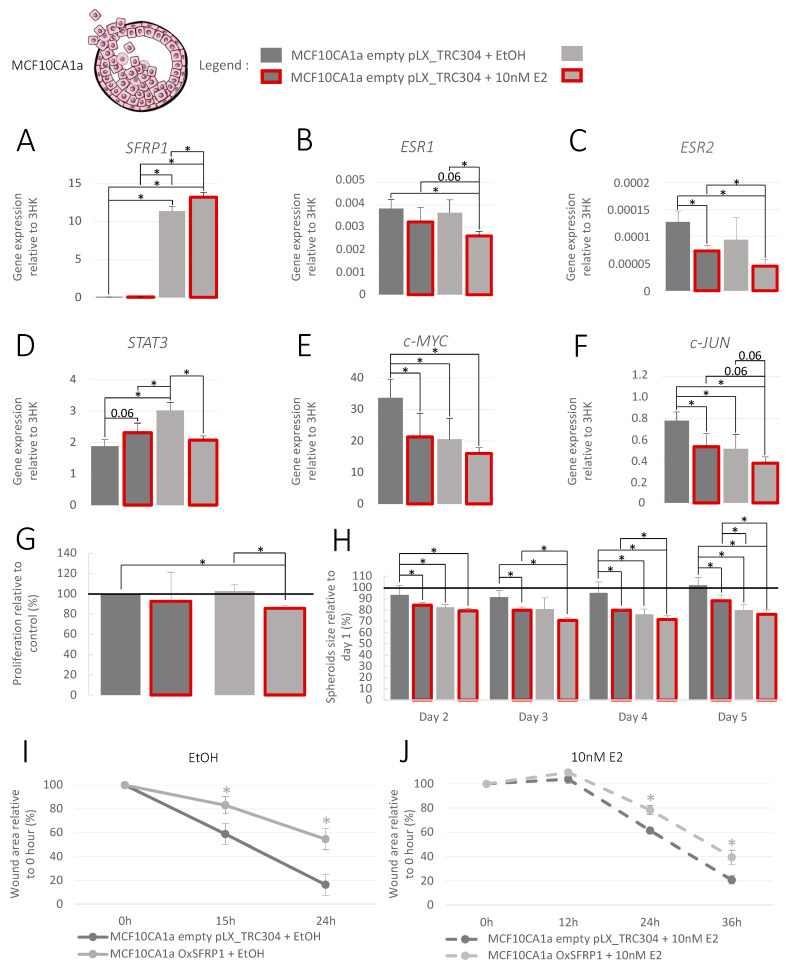
Increase in *SFRP1* expression in MCF10CA1a cell line. (**A**) *SFRP1*, (**B**) *ESR1*, (**C**) *ESR2*, (**D**) *STAT3*, (**E**) *c-MYC,* and (**F**) *c-JUN* expression relative to three housekeeping genes (HK) quantified by qPCR in MCF10CA1a OxSFRP1 and empty pLX_TRC304 negative control, in EtOH and E2 conditions. (**G**) Viability assay comparing MCF1CA1a empty pLX_TRC304 negative control to MCF10CA1a OxSFRP1 in both EtOH and E2 conditions. (**H**) Average spheroid size and regarding *SFRP1* modulation of expression in MCF10CA1a cell line, in both EtOH and E2 conditions. Wound-healing assay comparing MCF10CA1a empty pLX_TRC304 negative control to MCF10CA1a OxSFRP1 in (**I**) EtOH and (**J**) 10 nM E2 conditions. All experiments were performed in technical triplicate and repeated three times independently. * = *p*-value < 0.05.

**Figure 7 cancers-15-02251-f007:**
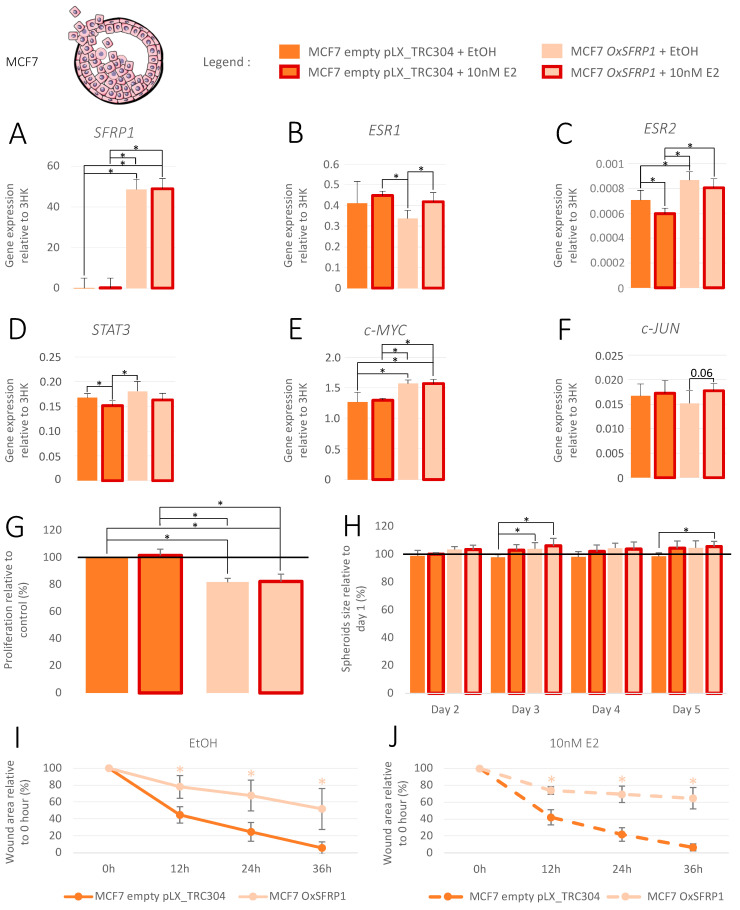
Increase in *SFRP1* expression in MCF7 cell line. (**A**) *SFRP1*, (**B**) *ESR1*, (**C**) *ESR2*, (**D**) *STAT3*, (**E**) *c-MYC,* and (**F**) *c-JUN* expression relative to three housekeeping genes (HK) quantified by qPCR in MCF7 OxSFRP1 and empty pLX_TRC304 negative control, in EtOH and E2 conditions. (**G**) Viability assay comparing MCF7 empty pLX_TRC304 negative control to MCF7 OxSFRP1 in both EtOH and E2 conditions. (**H**) Average spheroid size and regarding *SFRP1* modulation of expression in MCF7 cell line, in both EtOH and E2 conditions. Wound-healing assay comparing MCF7 empty pLX_TRC304 negative control to MCF7 OxSFRP1 in (**I**) EtOH and (**J**) 10 nM E2 conditions. All experiments were performed in technical triplicate and repeated three times independently. * = *p*-value < 0.05.

**Figure 8 cancers-15-02251-f008:**
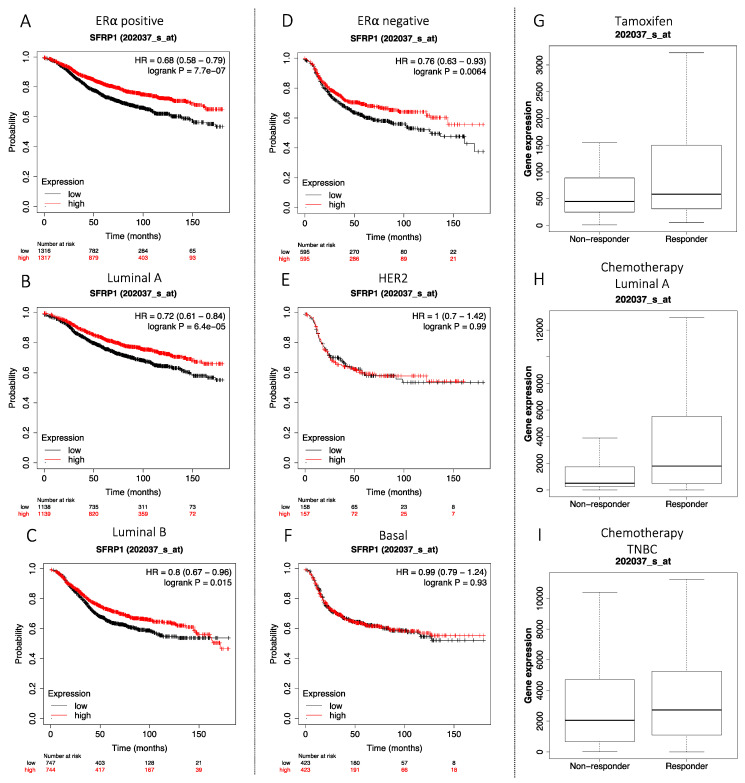
Association between *SFRP1* expression and breast cancer outcomes. Association between *SFRP1* expression and (**A**) ERα-positive, (**B**) ERα-negative, (**C**) luminal A, (**D**) luminal B, (**E**) HER2, and (**F**) basal breast cancer molecular subtypes and breast cancer disease-free survival. Breast cancer molecular subtypes were assessed by immunohistochemistry following the St. Gallen expert consensus. Kaplan–Meier curves were drawn online with https://kmplot.com/analysis/ (accessed on 15 April 2021). (**G**) *SFRP1* expression in breast cancer tumor samples from patients treated with tamoxifen who relapsed (non-responder) compared to those who did not relapse (responder) in the 5 years following diagnosis. *SFRP1* expression among patients with complete pathological response (responder) to chemotherapy compared to patients who did not respond (non-responder) in (**H**) luminal A and (**I**) TNBC patients. Breast cancer molecular subtypes were assessed by immunohistochemistry following the St. Gallen expert consensus. Boxplots were draw online with http://rocplot.org/site/treatment/ (accessed on 2 June 2021).

**Table 1 cancers-15-02251-t001:** Cell line characteristics and culture medium recipes.

Cell Lines	Type of Lesion	Molecular Subtype	Stade/Grade	Age (Years)	Culture Medium
MCF10A	Normal	NA	NA	36	- DMEM F12, - 5% horse serum - 1% penicillin–streptomycin mixture (5000 IU penicillin, 5000 μg/mL) - 1% HEPES - 10 µg/mL insulin - 20 ng/mL epidermal growth factor - 0.5 µg/mL hydrocortisone
MCF10AT1	ADH	NA	NA		
MCF10DCIS.com	DCIS	NA	NA		- DMEM F12 - 5% horse serum - 1% penicillin–streptomycin mixture (5000 IU penicillin, 5000 μg/mL) - 1% HEPES
MCF10CA1	IDC	Basal-like	NA		- DMEM F12 - 5% horse serum - 1% penicillin–streptomycin mixture (5000 IU penicillin, 5000 μg/mL) - 1% HEPES
MCF7	IDC	Luminal A	Stage 4	69	- DMEM F12 without phenol red - 5% fetal bovine serum - 1% penicillin–streptomycin mixture (5000 IU penicillin, 5000 μg/mL) - 13.4 mL sodium bicarbonate - 7.5 mL HEPES - 1 nM estradiol (E2)

Abbreviations: ADH—atypical ductal hyperplasia; DCIS—ductal carcinoma in situ, IDC—invasive ductal carcinoma; NA—not available.

## Data Availability

Publicly available datasets were analyzed in this study. This data can be found here: [https://tnmplot.com/ (accessed on 15 April 2021); http://gent2.appex.kr/gent2/ (accessed on 15 April 2021); https://xenabrowser.net/ (accessed on 18 September 2021); http://www.rocplot.org/ (accessed on 2 June 2021); https://precog.stanford.edu/ (accessed on 16 April 2021); https://kmplot.com/analysis/ (accessed on 15 April 2021)].

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
