# Peer review of "Role of Secreted Frizzled-Related Protein 1 in Early Breast Carcinogenesis and Breast Cancer Aggressiveness"

_cancers, 2023, doi:10.3390/cancers15082251_

Round 1

Reviewer 1 Report

The authors aim to establish a link between the increased risk of nulliparous women in developing estrogen-dependent breast cancer with the expression of SFRP1 in breast tissue. Analyses of data banks and studies with organoids from nulli- and mulitparous mice show differences in SFRP1 expression in breast cancers vs. normal breast tissue and differences in responses of tissues from nulli- vs multiparous mice to certain stimuli. Then the authors switch to the MCF10A  cell line to further analyze the importance of SFRP1 in the development for ERalpha-dependent tumors. This analyses encompass a large part of their study. However, as a triple-negative cell line MCF10A cell line may not be an appropriate cell line to analyze the effect between SFRP1 and ERalpha expression. Can the authors show that MCF10A cells can transform to ERalpha-dependent cancer cells? This would be essential to justify the use MCF10A for this particular study. Instead, the authors used a basal-like MCF10A-derived cancer cell line. To me, it does not make sense.

KD and overexpression studies with SFRP1 reveal contradictory results. The relationship between SFRP1 and ESR1 expression remains unclear.

Anti-estrogens should be used to clarify the role of ERalpha in the effect of SFRP1 on cell viability and migration.

The study also lacks protein expression analyses, particularly on SFRP1 and ERalpha.

Specific comments

Introduction

In the introduction, molecular and immunohistochemical (IHC) subtypes are mixed. Luminal subtypes belong to the molecular taxonomy, Her2-positive tumors and TNBCs to the IHC taxonomy. Later it is mentioned that luminal B is Her2-enriched. This is not correct. Luminal B and Her2-enriched are two different molecular subtypes. Please be precise. Also please make it clear that PAM50 is a diagnostic method, which has been established based on the molecular subtype distinction published years before.

M&M

Have the cell lines and sublines used be authenticated by SNP analysis?

Results

Fig. 1B, Suppl. Fig. 1D, Fig. 8: It is not clear whether “Her2” means Her-positive by IHC or Her2-enriched by molecular subtyping.

Fig. 1A, Suppl. Fig. 1A: How do the authors explain that SFRP1 expression is lower in metastatic vs. primary tumors while being higher in poorly differentiated tumors vs. well differentiated tumors?

lines 412: c-Myc and c-Jun are not only regulated by the Wnt pathway. The differences in c-Jun expression as found by the authors might be Wnt-independent. Further evidence should be provided before claiming that the differences in the responses of the organoids derived from nulli- and multiparous mice are Wnt-dependent.

3.3.1 The MCF10A cell line is an immortalized triple-negative breast cell line with low expression of ERalpha, which does show increased proliferation in response to estrogen  (e.g. Murata et al. Molecular Aspects of Alcohol and Nutrition 2016, Pages 315-324).  Another paper shows that MCF10A express as little ERalpha as the TNBC line MDA-MB-231 (Datta et al. Activity of Estrogen Receptor β Agonists in Therapy-Resistant Estrogen Receptor-Positive Breast Cancer, bioRxiv 2022). Hence, MCF10A is not an appropriate cell line to analyze the link between ERalpha and SFRP1.

lines 487-499, 572-574: The effect of SFRP1 KD on STAT3 mRNA is low. More interesting would have been to analyze the effect of this shRNA on STAT3 activity.

lines 612-615: The statement “Indeed, these results confirmed that the negative correlation between SFRP1 and ESR1 expression is associated with an increase in E2 sensitivity (Figure 4J) and induces the acquisition of tumoral properties such as an increase in cell viability (Figure 4G)” is not true. MCF10AT1 sh62170 showed a strong decrease in cell viability. Plus, since this subline showed a similar increase in ESR1 expression as the other MCF10AT1 and the MCF10A sublines, cell viability cannot be linked to the increased ESR1 expression. It seems to me that even the 2-3-fold increase of the low ESR1 expression does not make viability of MCF10A cells more dependent on ESR1.

Furthermore, have the authors tried to block ESR1 activity by anti-estrogens to check whether the biological effects of SFRP1 KD are ESR1-dependent?

lines 625-627: “Interestingly, the overexpression of SFRP1 in MCF10DCIS + EtOH induced a 1.5 ± 0.3-fold increase in ESR1 expression compared to MCF10DCIS empty pLX_TRC304 + EtOH”. This does not make sense, if there is an inverse relationship between SFRP1 and ESR1 expression.  The authors’ explanation “the existence of a differential role of SFRP1 according to breast cancer stage and molecular subtype.” is very vague. What happens when SFRP1 is overexpressed in MCF10A or MCF10AT1?  Does overexpression of SFRP1 in these cell lines also increase ESR1 expression.

line 114: typo “dependant”

The authors often use the word “misunderstood”. I think, in the context it is used, “not well understood” would be more appropiate.

Reviewer 2 Report

The Introduction is well written and provides a good background information.

Materials and methods are also well described.

The focus of the article is on ER-positive breast cancers because early on authors provide evidence for lower expression of SFRP1 in ER-positive patients, as compared to HER-2-overexpressing or the TNBC patients. While this makes sense in terms of authors’ work, I have difficulty comprehending that HER-2-overexpressing and TNBC, that are associated with increased metastases, have higher expression of SFRP1 which seems to be inhibit metastasis. This begs a question whether or not this factor is really associated with metastasis !!

In continuation of my above concern, it would be interesting to see if the public database analysis on this relative expression of SFRP1 actually corelates with SFRP1 levels in cell lines representing different breast cancers. In particular, it would be interesting to see expression levels in HER-2-overexpressing SKBR3 or the TNBC cells such as MDA-MB-231.

Other than the above mentioned concerns, the other Results are well described.

The conclusions section needs to be re-written to accurately and comprehensively cover all the major findings from the study.

Round 2

Reviewer 1 Report

The authors have addressed my comments satisfactorily. Therefore, I recommend publication of the manuscript in its revised version.

Reviewer 2 Report

Thanks for addressing all of my concerns !